# Quantifying radiative effects of light–absorbing particles deposition on snow at the SnowMIP sites

Enrico Zorzetto[1,2,3], Paul Ginoux[4], Sergey Malyshev[4], and Elena Shevliakova[4]

[1]Hantush-Deju National Center for Hydrologic Innovation, New Mexico Institute of Mining and Technology, Socorro, NM, USA
[2]Earth and Environmental Science Department, New Mexico Institute of Mining and Technology, Socorro, NM, USA
[3]Atmospheric and Oceanic Sciences Program, Princeton University, Princeton, NJ, USA
[4]NOAA OAR Geophysical Fluid Dynamics Laboratory, Princeton, NJ, USA.

**Correspondence:** Enrico Zorzetto (enrico.zorzetto@nmt.edu)

**Abstract.** The deposition of light-absorbing particles (LAPs) leads to a decrease of surface albedo over snow covered surfaces. This effect, by increasing the energy absorbed by the snowpack, enhances snow melt and accelerates snow aging, process which in turn is responsible for further decreasing the snow albedo. Capturing this combined process is important in land surface modelling, as the change in surface reflectivity connected with the deposition of LAPs can modulate time and magnitude of snowmelt and runoff. These processes impact regional water resources, and can also lead to relevant feedbacks to the global climate system. We have recently developed a new numerical snowpack model for the GFDL land model (A Global Land Snow Scheme, or GLASS). GLASS provides a detailed description of snow mass and energy balance, as well as the evolution of snow microphysical properties (grain shape and size). We now extend this model to account for the presence of light-absorbing impurities, modelling their dry and wet deposition in the snowpack, the evolution of their vertical distribution in the snow due to precipitation and snow melt, and the effect of their concentration on snow optical properties. To test the effects of the resulting snow scheme, we force the GFDL land model with deposition of black carbon, mineral dust and organic carbon obtained from a general circulation model (GFDL AM4.0). We evaluate the new model configuration at a set of instrumented sites, including an alpine site (Col de Porte, France) where in-situ observations of snow (including spectral measurements of snow reflectivity and concentration of LAPs) allow for a comprehensive model evaluation. For the Col de Porte site, we show that GLASS reproduces the observed magnitudes of impurities concentration in the snowpack throughout a winter season. The seasonal evolution of the snow optical diameter is also qualitatively reproduced by the model, although the increase in snow grain diameter during the melt season appears to be underestimated. For a set of instrumented sites spanning a range of climates and LAP deposition rates (the 'SnowMIP' sites) we then evaluate the number of snow-days lost due to the deposition of dust and carbonaceous aerosols. We find that this loss ranges between 5 and 24 days depending on the site. The resulting snow model with LAP-aware snow reflectivity show a good agreement with measurements of broadband albedo and seasonal SWE over the study sites.

# 1 Introduction

The deposition of light absorbing particles (LAPs) on snow is known to reduce its surface reflectivity, in turn accelerating snow melt and snow aging through enhanced metamorphism (Warren, 1984; Doherty et al., 2010; Dumont et al., 2014; Hadley and Kirchstetter, 2012; Skiles et al., 2018; Sarangi et al., 2020). The impact of LAPs has been shown to lead to relevant impacts in the water cycle over extended regions of the world, such as the western US (Painter et al., 2012), the Alps (Di Mauro et al., 2015) and Pyrenees (Réveillet et al., 2022) and high mountain Asia (Ackroyd et al., 2021; He et al., 2018a), among other regions. The effects of the concentration of LAPs on snow optical properties is very nonlinear: While the deposition of LAPs directly affects snow albedo in the visible range, it additionally produces indirect effects. As the energy absorbed by snow increases, the evolution of snow grain size and shape with snow aging accelerates, driven both by thermal gradients and wet processes (Hadley and Kirchstetter, 2012). In turn, these metamorphic changes on snow grain properties modulate snow albedo, not only in the visible range but also in the near-infrared part of the spectrum (Skiles et al., 2018), and can enhance the surface albedo feedback, especially during spring (Huang et al., 2022). Accounting for LAP deposition and its effects on snow albedo is thus important in numerical snow schemes which are routinely used in hydrological and land surface models, as these processes impact the spatial variability of snow cover and can lead to long-term changes in snow at the ground. Furthermore, these changes in snow cover exert control both on surface and sub-surface hydrology (through changes in snow melt timing) and on land surface-atmosphere interactions, through changes in land surface temperature. Accurate snow predictions are thus of paramount importance for Earth System Models, especially as snow water equivalent and snow extent have both been shown to decline both in historical observations (Estilow et al., 2015; Kunkel et al., 2016) and in climate projections (Mudryk et al., 2020).

Multiple LAP species can affect snow albedo. Common species of LAPs found in snow include mineral dust (Painter et al., 2007; Sarangi et al., 2020), black carbon (Flanner et al., 2007; Réveillet et al., 2022; Flanner et al., 2012), and organic carbon (resulting from both natural and anthropogenic combustion processes), volcanic ash, and other biological elements such as algae (Cook et al., 2017). LAPs can be added to snow by wet deposition (i.e., LAPs contained in liquid or frozen precipitation) and by dry deposition, whereby LAPs are deposited on snow by gravitational settling or turbulent/laminar exchange with the atmosphere. Once deposited, the concentration of LAPs in a snow layer evolves as a result of snow melt and sublimation (Conway et al., 1996). In the case of sublimation, the concentration of LAPs near the snowpack surface increases due to the net loss of ice to the atmosphere. On the other hand, snow melt can remove LAPs from the snow through a phenomenon referred to as *scavenging*, depending on whether the particles are hydrophilic or hydrophobic. It has been observed (Sterle et al., 2013) that black carbon and dust tend to be retained in the snowpack even during the ablation season, leading to increased LAP concentrations in the uppermost layers following snow melt.

The most abundant LAP by mass is mineral dust, which generally originates from deserts and other poorly vegetated and dry regions due to wind-driven emission. Light absorptive properties of mineral dust can vary greatly, and are primarily controlled by its iron content. Black carbon has the largest absorption per unit mass in the visible wavelength compared to organic carbon and other common LAPs.

In recent years, the growing recognition of i) the importance of LAPs in modulating snow processes, and ii) the interaction of these effects with the climate system and land hydrology has led to multiple modelling efforts focused on LAPs. Radiative transfer numerical models have been develop to represent the effect of LAPs in snow (Wiscombe and Warren, 1980; Liou et al., 2014; Flanner et al., 2007; Aoki et al., 2011; Libois et al., 2013; He et al., 2019), including simplified parameterizations tailored to global circulation models (Dang et al., 2015; He et al., 2018b), in which land surface schemes routinely employ a coarse 2-band representation of the solar spectrum. He et al. (2018a) and He et al. (2019) carried out an extensive evaluation of the radiative effects of LAPs in snow, focusing on the role of internal vs. external mixing state, and quantified the magnitude of their effects for snow grains of different shapes. However, all radiative transfer models rely on the knowledge of LAP concentrations and snow properties (e.g., snow optical diameter and grain shape) as well as the distribution of these properties within the snowpack. This presents a challenge, as most snow schemes employed in global Earth System models have a simplified representation of snowpack properties and their vertical distribution.

Detailed snow models accounting for the LAP deposition process have been developed. For instance, Tuzet et al. (2017, 2020) extended the high-detail snow model CROCUS (Vionnet et al., 2012) to include LAP processes, and used to evaluate the effect of LAPs over instrumented sites in the French Alps. Similarly, (Skiles and Painter, 2019) employed the radiative transfer model SNICAR (Flanner and Zender, 2005) to quantify LAPs effect in the snow model SNOWPACK (Lehning et al., 2002). However, generally highly detailed snow models are used in local or regional studies due to their complexity and computational cost. In comparison to these local studies, several Earth System Models still employ simple representation of snow processes such as LAP deposition, snow metamorphism, and a coarse vertical representation of the snowpack which does not allow to adequately characterize the vertical heterogeneity of the snowpack. These large differences in complexity and detail of snow schemes used in different ESMs have lead to large inter-model differences being observed in previous model intercomparison efforts (Nijssen et al., 2003; Krinner et al., 2018; Menard et al., 2021) and are thought to contribute to the large spread still observed between model estimates of the surface albedo feedback (Flanner et al., 2011; Qu and Hall, 2014).

Therefore, understanding to what extent the representation of LAP-on-snow processes contributes to the uncertainty in snow predictions from regional and global modelling efforts is a key scientific question which in the last decade has received increasing attention in the Earth System Modelling community (Qian et al., 2015; Réveillet et al., 2022; Hao et al., 2023b).

The effect of LAPs has for example been implemented in the Community Land Model (CLM) using the SNICAR (Snow, Ice, and Aerosol Radiation Model) developed by Flanner and Zender (2005); Flanner et al. (2009), finding that the presence of LAPs leads to increased surface temperature and to relevant feedbacks in a global circulation model. More recently, a SNICAR-based parameterization for LAPs effect on snow optical properties was implemented in the DOE E3SM model (Golaz et al., 2022), and used to test the representation of snow processes over the Tibetan plateau (Hao et al., 2022) and over the Western United States (Hao et al., 2023a).

Despite the recent progress, the global distribution of LAPs effects on snow remains characterized by large uncertainties, which include i) the magnitude of this effect, ii) the relative contributions of different LAP species, and iii) the interactions of LAPs-driven melt with snow metamorphic processes. To reduce this uncertainty, here we analyze the effect of LAPs at instrumented sites using a recent detailed snow scheme (Global Land Snow Scheme, GLASS) developed for Earth System

Model simulations. In a companion manuscript (Zorzetto et al., 2024), we have presented this new snow scheme and discussed its implementation in the GFDL Earth System Model. GLASS includes a refined vertical structure of the snowpack and an explicit description of snow metamorphism. GLASS is based on an implicit numerical solution of mass and energy balance for the snowpack, so that the model can be effectively employed in global-scale simulations of the land-atmosphere coupled system. In this manuscript, we extend GLASS to include the effect of light absorbing impurities, including their wet and dry deposition, their mass balance within the snowpack, and their effects on snow optical properties, accounting for predicted snow microphysical structure (snow grain shape and optical diameter).

Harnessing this new model, here we focus on responding to two questions: How do LAPs affect snow melt in the spring? and, is the modelled snowpack in agreement with observations (bulk snow properties, grain properties, and LAP concentration near the snow surface?). While GLASS is designed for global applications, we focus our analysis on 10 "SnowMIP" sites (Ménard et al., 2019), as i) all these sites include high-quality forcing data and validation data, which allow to reduce uncertainty compared to large-scale studies and are thus invaluable to evaluate model performance, and ii) offer a perfect opportunity to quantify LAP-driven snow melt for a set of site spanning a wide range of climate and terrain conditions, which can be used to further constrain large scale studies. Furthermore, this forcing and validation dataset was used in a recent SnowMIP ESM comparison effort (Krinner et al., 2018; Menard et al., 2021).

The paper is organized as follows: We first provide an overview of the GFDL land model and of the GLASS snow scheme. We then describe the new treatment of LAPs mass balance within the snowpack, and the revised snow albedo model used to capture LAP-driven snow darkening processes. The experimental setup is then discussed, along with the datasets used for forcing the model and for validation. A particular focus of the discussion is the Col de Porte site (France), where both measurements of snow bulk properties and spectral measurements were carried out for the 2013-2014 snow season which allow us to evaluate model predictions of snow optical diameter and of LAP concentration at the snowpack surface. We then discuss the implications of our findings for land surface simulations over continental domains and in coupled land-atmosphere simulations.

## 2 Model Description

### 2.1 GFDL land model

The land model LM4.1 (Shevliakova et al., 2024) is the land component of the GFDL ESM4.1 (Dunne et al., 2020). The description of water and energy balance at the land surface and in subsurface soil layers is based on the Land Dynamics scheme (Milly et al., 2014). The land domain is modelled in a grid of cells composed of sub-units, termed tiles, which represent homogeneous areas of either soil, lake or glacier. In the present work we employ a "*point model*" version of LM4.1, whereby we assume that a single land model tile represent the soil-snow-atmosphere continuum at each site. Here we use the same physics time step routinely used in global-scale simulations (30 minutes). Soil is modelled as a multi-layer medium coupled with snow and atmosphere above, with full representation of mass balance of liquid and frozen water, and vertical diffusion of heat. Exchanges of water and energy between land and atmosphere are computed according to the *Monin-Obukhov* similarity

theory (Garratt, 1994). In LM4.1, vegetation is represented by a set of plant cohorts which evolve dynamically. Multi-layer vegetation canopies interact with the surface via multiple processes, including turbulent exchange of mass and energy, the transfer of longwave and shortwave radiation, and the interception of liquid and frozen precipitation. For additional details, the reader is referred to Shevliakova et al. (2024). In this application, we have decided to limit our analysis to sites with little to no vegetation in order to focus our attention on snow and LAP deposition processes. Therefore, vegetation is turned off and

canopy layers are not present in the model simulations, following the experimental setup used in (Zorzetto et al., 2024) for sites with little or no vegetation.

## 2.2   Snow model

The physical description of snow processes is based on a snow scheme recently developed for LM4.1 (the Global LAnd Surface Scheme, or GLASS, described by Zorzetto et al. (2024)). In GLASS, the snowpack is modelled as a 1D multi-layer medium.

Each layer $k$ ($k = 1, \ldots, n_L$ from the top), is characterized by ice ($w_{s,k}$) and liquid ($w_{s,k}$) contents (in $\mathrm{kg\,m^{-2}}$), temperature, density (i.e., layer thickness $\Delta z_k$), snow age, and a set of variables describing the snow microphysical structure, which evolve based on dry and wet metamorphic processes, snow compaction, and wind drift effects. These variables are snow optical diameter ($d_{opt,k}$), snow grain sphericity ($s_{p,k}$), and grain dendricity ($\delta_k$). Snow metamorphism caused by dry processes driven by temperature gradients is described according to (Flanner and Zender, 2006), while wet processes are modelled following

the parameterization by (Brun et al., 1992).

    The evolution of snow grain shape is described based on the prognostic equations for snow grain dendricity and sphericity. These are computed following the parameterization used in CROCUS (Brun et al., 1992; Vionnet et al., 2012; Carmagnola et al., 2014). In GLASS, all three microphysical properties ($d_{opt,k}$, $s_{p,k}$, and $\delta_k$) are prognostic variables. Wind drift and snow compaction are also accounted for in the model following the approach by (Vionnet et al., 2012) and contribute to the evolution

of snow density and grain size size and shape. Snow albedo and diffusion of shortwave radiation within the snowpack are parameterized based on snow grain size and shape. The vertical structure of the snowpack consists of a dynamic number $n_L$ of snow layers. New layers are created on top of the existing snowpack following snowfall events of large enough magnitude, so that the vertical layering structure preserves snow physical properties in each layer. Depending on snowfall rate, up to 5 new snow layers can be created during a single model time step. The vertical layers are also updated based on computational

considerations. At each time step, the snow vertical structure is compared to an optimal vertical discretization defined for each given snow depth. If the layers are too coarse or too thin for a given snow depth, the layers undergo splitting or merging. In the current configuration, the optimal thickness of the uppermost snow layer is set to $3\,\mathrm{cm}$ , and each snow layer optimal thickness is set to 1.5 that of the layer immediately above it, so that in general the model allows for thinner layers closer to the surface, while within the snowpack layer thickness increases with depth. The model does not prescribe a maximum number of layers,

while if snow is present the minimum number of layers is 3, as required for numerical solution of mass and energy vertical balance equations. These operations are designed in order to strike a trade-off between computational cost and vertical detail, to satisfy requirements of numerical efficiency (to avoid too large number of layers) and to ensure a proper description of the snowpack vertical structure (too coarse a vertical discretization would hinder the representation of some physical processes,

such as the vertical heat diffusion). If two snow layers are characterized by values of density, optical diameter or impurities content which are too different, merging of the two layers is not permitted in order to preserve the vertical heterogeneity of the snowpack.

At each model time step, the full energy and water mass balance of the snowpack is solved using an implicit numerical formulation, which is required for land-atmosphere coupled model runs with relatively coarse time step (30 min) for which the GLASS was designed. After performing the vertical heat balance of the snowpack, the temperature change and change of phase is evaluated for all snow layers. The vertical balance of liquid water is then evaluated throughout the snowpack, with liquid precipitation providing the upper boundary condition. The snow density is used to evaluate the pore space available for liquid water in each snow layer. Following Vionnet et al. (2012), the liquid water holding capacity for a snow layer $k$ with thickness $\Delta z_k$ and local solid-phase density $\rho_{s,k}$ is given by

$$W_{liq,max,k} = 0.05\rho_w\Delta z_k \left(1 - \frac{\rho_{s,k}}{\rho_i}\right) \tag{1}$$

with $\rho_i$ the density of ice, and $\rho_w$ that of liquid water. For a more detailed description, see Zorzetto et al. (2024).

### 2.3 Tracer deposition in the snowpack

In this work GLASS was updated to include the mass balance of multiple LAP species. In addition to the set of variables mentioned in Section 2.2, in this revised version each snow layer $k$ is further characterized by the mass of each LAP specie both internally mixed (IM) and externally mixed (EM) within the snow. These quantities ($w_{IM,i,k}$ and $w_{EM,i,k}$) in [$\mathrm{kg\,m^{-2}}$] are tracked for each tracer species $i$, so that in our current application we have 6 types of LAP in each layer (IM and EM, for each species: BC, OC and MD). The model considers a single LAP size distribution for mineral dust, without tracking separately dust particles of different sizes. Similarly, the current model considers a single BC species and does not distinguish between hydrophobic and hydrophilic components. This is a limitation as hydrophobic and hydrophilic BC species have different optical properties and scavenging coefficients (Flanner et al., 2007). However, we note that the model can in principle be extended to track multiple BC species (or multiple dust size bins) as long as their optical properties and scavenging coefficients are known. A conceptual representation of the new processes included in GLASS are illustrated in Figure 1. The mass of LAPs added to the snowpack by dry deposition is assumed to be externally mixed (EM), while LAPs deposited as wet deposition either due to liquid or frozen precipitation contribute to IM LAPs within the snowpack. In the model, the state of mixing of LAPs within the snow (IM or EM) does not change as a consequence of melt and freeze cycles occurring in a snow layer.

The dry flux $D_{c,i}$ (in $\mathrm{mg/m^2/s}$) for each tracer specie (i=BC, MD, OC) is deposited in the uppermost snow layer. While in general wind pumping can redistribute dry deposited LAPs within the snow, this process is limited to thicknesses smaller than $10\,\mathrm{mm}$ so that here we decided to assign all deposition to the uppermost snow layer (Clifton et al., 2008; Tuzet et al., 2017). At each model time step, we assume dry deposition is equal to the monthly average value from the forcing dataset.

The wet deposition flux in our forcing dataset was provided as monthly average. Since we force the snow model with in-situ observations, we adopt the following strategy to estimate wet deposition fluxes for each snowfall or rainfall event: We first

compute the monthly average concentration of each LAP specie in the precipitation (as the mass ratio of monthly average wet deposition to monthly precipitation, in [ppm]). We then assign in each model time step the total amount of wet-deposited LAPs as proportional to the rainfall and/or snowfall rate for that time step, so that the flux of tracer $i$ due to liquid and solid precipitation is respectively $c_{l,i}f_l$ and $c_{s,i}f_s$. Note that based on this procedure the mixing ratio of LAPs in precipitation is constant during each month, and exhibits step changes across months. For the purpose of this study, we assume that rainfall and snowfall carry the same concentration of LAPs, and neglect any possible dependence of deposition fluxes e.g., on precipitation intensity.

In the case of large enough snowfall, the snow model creates a set of new snow layers on top of the existing snowpack. The newly deposited LAP mass is stored in these layers. In case of small snowfall events and snow already present at the ground, instead of creating new snow layers the model adds the fresh snow to the existing upper layer of the snowpack. In that case, the deposited LAPs are also added to the existing mass of each species in that layer.

## 2.4 LAPs in the snow model

The land model solves the energy balance at the ground using an implicit time stepping scheme. From this step, the snow temperature profile is updated and an estimate of the mass available for melt or freeze within the snowpack is computed. Sublimation leads to a thinning of the uppermost snow layer, or to its complete disappearance and to mass being removed from the underlying layers. In both cases, any existing mass of LAPs present in these layers is conserved. Thus, in general sublimation leads to an increase in concentration of LAPs at the top of the snowpack. If the entire snowpack disappears due to sublimation, any existing mass of LAPs is lost. Sublimation does not lead to changes in the status of LAPs (internally vs externally mixed). Then, melt and freeze are applied to each layer of the snowpack, if the thermodynamic conditions and availability of liquid water or ice require it.

The mass of each LAP species is conserved by snowpack relayering operations. If a snow layer is split in two, the amount of each LAP species present in the layer is assigned to the new layers proportionally to their snow water masses. Similarly, in the event of two snow layers being merged, the LAP mass in the newly formed layer will simply be the sum of that in the original layers, for each LAP species and LAP mixing state. When a snow layer disappears as a consequence of melt or sublimation, the entire LAP content of the layer accumulates in the layer immediately underneath, if any, or else is lost from the snowpack.

## 2.5 Snow melt

When a snow layer completely melts, the mass of LAPs present in that layer is deposited to the layer underneath. When there is liquid water flow between a layer and the one underneath, a part of the LAPs present in the upper layer is scavenged by the water flow. The fraction of LAPs removed is based on a scavenging coefficient. Any LAPs carried by water flowing from snowpack to the substrate (glacier, lake or soil) are lost. For snow layer $k$ and LAP species $i$ the vertical flux (between layers $k$ and the underlying $k+1$) of LAPs is given by:

$$M_{scav,i,k} = q_{L,i,k} C_{scav,i} \frac{w_{c,i,k}}{SWE_k} \tag{2}$$

As pointed out by Sterle et al. (2013), the magnitude of scavenging for dust and BC is highly uncertain, and appears to be low due to the tendency of these LAPs to remain in the snowpack throughout the ablation processes. Similar to previous studies (Tuzet et al., 2017; Ga Chan et al., 2022), here we set the scavenging coefficient to 0 for mineral dust (with particles generally too large for scavenging to occur) and organic carbon, and to 0.2 for black carbon, as suggested by (Flanner et al., 2007).

## 2.6 Snow albedo parameterization

In this work, the snow surface albedo is computed based on snow properties (optical diameter and shape) and on the concentration of LAPs near the snowpack surface. In this section, snow properties are averaged over a near-surface layer of thickness set equal to up to $3\,\mathrm{cm}$. If the snowpack is thinner than $3\,\mathrm{cm}$, the near-surface layer includes the entire snow depth. If the upper snow layers are thinner than $3\,\mathrm{cm}$, the near-surface snow properties are computed as weighted average across snow layers of the snow properties in each layer, up to a $3\,\mathrm{cm}$ depth. In GLASS, the shortwave radiative balance is resolved for two bands, visible (VIS) and near infrared (NIR), separated at $700\,\mathrm{nm}$. Based on the work of Dang et al. (2015) and He et al. (2018b), in GLASS the snow surface albedo for each band ($b = VIS$ or $b = NIR$) is expressed as a function of snow grain effective radius

$$\alpha_b = b_0 \left(b, \delta_{ns}, s_{p,ns}\right) + b_1 \left(b, \delta_{ns}, s_{p,ns}\right) R_n + b_2 \left(b, \delta_{ns}, s_{p,ns}\right) R_n^2 - \Delta\alpha_b \tag{3}$$

with

$$R_n = \log_{10}\left(\frac{R_e \phi_b\left(\mu\right)}{R_0}\right) \tag{4}$$

here $R_e$ represents the snow effective radius, defined as $R_e = 3V_s/(4A_s)$ with $V_s$ the snow grain volume, and $A_s$ its surface area projection. $R_e$ corresponds to $d_{opt,ns}/2$. The reference radius is $R_0 = 100\,\mu\mathrm{m}$. Here the optical diameter $d_{opt,ns}$ as well as the snow grain shape parameters $\delta_{ns}$ and $s_{p,ns}$ are averaged over the near surface layer of the snowpack, with thickness up to the upper $3\,\mathrm{cm}$. $\Delta\alpha_b$ represents the decrease in snow surface albedo due to the effect of LAP concentration in snow. This effect is present only for the visible band ($b = VIS$) as described in Section 2.7, while $\Delta\alpha_{NIR} = 0$. For direct radiation, we account for the direction of the incident radiation beam by computing an effective snow grain radius as proposed by Marshall (1989). This is achieved by the factor $\phi_b\left(\mu\right)$ in eq. (4) which modifies the effective radius

$$\phi_b\left(\mu\right) = \left(1 + a_{\theta,b}\Delta\mu\right)^2 \tag{5}$$

where $a_{\theta,b} = 0.781$ for visible band ($b = VIS$) and $a_{\theta,b} = 0.791$ for near infrared radiation ($b = NIR$). Here $\Delta\mu = \mu - \mu_D$, with $\mu = \cos\theta$ is the cosine of the solar zenith angle. In the case of diffuse radiation, we have $\mu = \mu_D = 0.65$ ($\theta = 49.5°$). The snow albedo parameterization used here explicitly accounts for the effects of snow grain size (through the snow grain effective radius) and shape on its optical properties. He et al. (2018b) introduced eq. (3) and provided the set of parameters $b_0$, $b_1$, and $b_2$ tabulated for four different snow grain shapes (sphere, spheroid, hexagon, and Koch snowflake). In GLASS, snow microphysics in each snow layer is parameterized by two parameters (snow sphericity and dendricity) which evolve in

time due to the combined effect of dry and wet snow metamorphic processes, as well as due to wind effects (Zorzetto et al., 2024). The coefficients used in eq. (3) are selected at each time step based on snow shape properties in the near-surface snow layer: High-dendricity snow ($\delta_{ns} > 0.5$) is idealized as a collection of Koch snowflakes. Snow with lower dendricity parameter is considered as a collection of spheres (if sphericity parameter $s_{p,ns} > 0.8$), spheroids (if $0.8 > s_{p,ns} > 0.2$), or hexagonal crystals (if $s_{p,ns} < 0.2$).

For thick enough snowpack (snow depth $h_s > 0.02\,\mathrm{m}$), solar radiation penetrates within the snowpack and absorbed radiation is distributed exponentially

$$Q_s\left(z\right) = \sum_{b=1}^{2}\left(1 - \alpha_b\right)R_{s,b}e^{-\beta_b z} \tag{6}$$

where for each band $b$ $R_{s,b}$ is the downward shortwave radiation at the surface, and $\beta_b$ describes the penetration of light within the snowpack. The extinction coefficients for visible and near infrared light are estimated as in Jordan (1991) and Shrestha et al. (2010): $\beta_{NIR} = 400$, and $\beta_{VIS} = 0.003759\,\rho\,d_{opt}^{-0.5}$, with density and optical diameter averaged over the near-surface layer of the snowpack, up to a maximum depth of $3\,\mathrm{cm}$.

Note that the albedo parameterization employed here based on the work of Dang et al. (2015) and He et al. (2018b) is derived for a semi-infinite snowpack, and thus in general can lead to biased albedo estimate for thin snowpack. In the case of this snowpack, the model computes a fractional snow cover $f_{snow}$ based on snow depth as follows

$$f_{snow} = \frac{h_s}{h_s + h_{s,c}} \tag{7}$$

with $h_s$ the snowpack depth, and $h_{s,c} = 0.0167\,\mathrm{m}$. In case of fractional snow cover, surface albedo is computed as a weighted spatial average of snow and snow-free substrate optical properties.

## 2.7  Effect of LAPs on predicted albedo

The effect of LAPs is accounted for using the parameterization by He et al. (2018b), in which the albedo reduction in the visible range is obtained as

$$\Delta\alpha_{VIS} = d_{0,b}\left(\delta_{ns}, s_{p,ns}\right)\left(C_{eq,bc}\right)^{k} \tag{8}$$

with

$$k = d_{1,b}\left(\delta_{ns}, s_{p,ns}\right)\left(\frac{R_e}{R_0}\right)^{d_2} \tag{9}$$

Here the parameters $d_{0,b}\left(\delta_{ns}, s_{p,ns}\right)$ and $d_{1,b}\left(\delta_{ns}, s_{p,ns}\right)$ are given by He et al. (2018b) as a function of spectral band (visible and near-infrared bands are used here) and as a function of snow grain shape (sphere, spheroid, hexagon or Koch snowflake). As done for the parameters of eq. (3), we account for the effect of snow grain shape on visible albedo reduction due to LAPs. Using the prognostic equation describing snow dendricity and sphericity, we evaluate the parameter in eq. 8 based on the value provided by He et al. (2018b).

**Table 1.** Characteristics of the experimental sites used for model validation.

| Station | Name | Years | Latitude | Longitude | Elevation [m] | Climate type |
|---|---|---|---|---|---|---|
| Col de Porte, FR | *cdp* | 1994-2014 | 45.30 N | 5.77 E | 1325 | Alpine |
| Reynolds Mountain East., USA | *rme* | 1988-2008 | 43.06 N | 116.75 W | 2060 | Alpine |
| Senator Beck, USA | *snb* | 2005-2015 | 37.91 N | 107.73 W | 3714 | Alpine |
| Swamp Angel, USA | *swa* | 2005-2015 | 37.91 N | 107.71 W | 3371 | Alpine |
| Weissfluhjoch, CH | *wfj* | 1996-2016 | 46.83 N | 9.81 E | 2540 | Alpine |
| Sapporo, JP | *sap* | 2005-2015 | 43.08 N | 141.34 E | 15 | Maritime |
| Sodankyla, FI | *sod* | 2007-2014 | 67.37 N | 26.63 E | 179 | Arctic |

To account for the radiative effects of multiple tracer species, we compute a BC-equivalent concentration as done by Ga Chan et al. (2022):

$$c_{eq,BC} = c_{BC} + c_{MD}\frac{\sigma_{abs,MD}}{\sigma_{abs,BC}} + c_{OC}\frac{\sigma_{abs,OC}}{\sigma_{abs,BC}} \tag{10}$$

Where $\sigma_{abs,t}$ is the absorption cross section of tracer $t$. These are set to 7330 $\mathrm{m}^2\mathrm{kg}^{-1}$ for BC, 67.8 $\mathrm{m}^2\mathrm{kg}^{-1}$ for MD, and 122 $\mathrm{m}^2\mathrm{kg}^{-1}$ for OC, values used by Ga Chan et al. (2022). All these quantities are averaged over the near-surface layer of the snowpack.

## 3   Experimental setup

### 3.1   Forcing and Validation Data

In this study, we use the land model driven by prescribed high-frequency meteorological data, including short-wave and long-wave downward radiation, rainfall and snowfall rates, air temperature, pressure, specific humidity, and wind speed. We first run a 100-year model spinup cycling through forcing data from the Global Soil Wetness Project Phase 3 (GSWP3) (1980-1990) with site-specific correction described in (Ménard et al., 2019). Then, we run the historical 1980-1996 years again using GSWP3 corrected data. Finally, we run the model for the entire duration of the in-situ meteorological observations available at

each site (e.g., 1996-2014 for the Col de Porte site). As done in Zorzetto et al. (2024), after spinning up the model at each site, we have evaluated the memory of the soil temperature and ice content and found that equilibrium is reached in about 30 model years. The characteristics of the test sites are summarized in Table 1.

The dataset used for testing model performance at Col de Porte is described in Dumont et al. (2017); Morin et al. (2012); Lejeune et al. (2019) and was previously used to evaluate the effect of LAPs in snow schemes Tuzet et al. (2017); Ga Chan

et al. (2022). The data collected at this site is extensive and include long series of snow depth, runoff and temperature, as well as daily average snow broadband albedo. In this study, we compare daily averages of the model output with the daily averages of observational data.

Additionally, spectral measurements were carried out in the snow year 2013-2014 at the Col de Porte site (Dumont et al., 2017), which were then used to estimate snow specific surface area (SSA) and concentration of LAPs. Dumont et al. (2017) used a theoretical spectral model to infer snow surface properties from a set of observed spectra. To quantify the uncertainty and artifacts in the measurements, a scaling factor $a$ was used to relate theoretical and observed spectra. To quantify the uncertainty of these retrievals, here we use the snow surface properties computed by Dumont et al. (2017) for three values of this scaling factor, corresponding to the $25^{th}$ ($a = 0.920$), $50^{th}$ ($a = 0.943$), and $^{75th}$ ($a = 0.964$) quantiles of its distribution.

In addition, for validation over the Senator Beck Basin (*snb*) site, we employ a dataset of dust-in-snow observations collected by Skiles and Painter (2015, 2017). These include end-of-year concentrations of dust within the snow for the years 2005-2012 (Skiles and Painter, 2015), and a sequence of measurements characterizing the seasonal evolution of MD and BC concentration in snowpack for the year 2013 (Skiles and Painter, 2017). Measured concentration values in this dataset correspond to average concentrations within the uppermost 30 cm of the snowpack. In the following, we compare these values to average modelled concentrations within the entire snowpack, and to the predicted concentration values over the snow near-surface layer.

## 3.2 LAP deposition data

For testing the new snow scheme, the land model is driven by aerosol deposition fluxes obtained by a fully coupled simulation with the GFDL AM4.0 and LM4.0 model (Zhao et al., 2018a, b). In the following we give a brief description of the model, and refer the reader to (Ga Chan et al., 2022) for an extensive description of the experimental setup.

AM4.0 tracks five types of aerosol, among which are the tracers used in this study (BC, MD and OC). Aerosols are characterized by a log-normal size distribution, except in the case of dust for which 5 size bins are used, with characteristic particle radii ranging from $0.1$ to $10\ \mu\mathrm{m}$. For each tracer species, the model simulates sources at the surface, atmospheric transport by advection, turbulent diffusion and convection, and deposition by both wet and dry processes. The lifting of dust is computed using the model by Ginoux et al. (2001), employing an empirical threshold for wind erosion, and using land cover data from CMIP6 forcing. Natural and anthropogenic sources of BC and OC were also obtained from CMIP6 emission data. The wet deposition process includes both effects of condensation within clouds, and below-cloud scavenging of aerosols by precipitation. The dry deposition of aerosols is driven by gravitational settling and turbulent exchange in the atmospheric boundary layer.

## 4 Results

### 4.1 Predictions for the Col de Porte site

We start by evaluating the model results at the Col de Porte (*cdp*) site for the 2013-2014 snow season. This is a useful benchmark, as i) the same site and season were used in previous studies making this a useful case study for model intercomparison (Tuzet et al., 2017; Ga Chan et al., 2022), and ii) at this site the measurements by Dumont et al. (2017) allow to evaluate not only snow bulk properties but also surface snow properties, namely snow optical diameter and concentration of impurities.

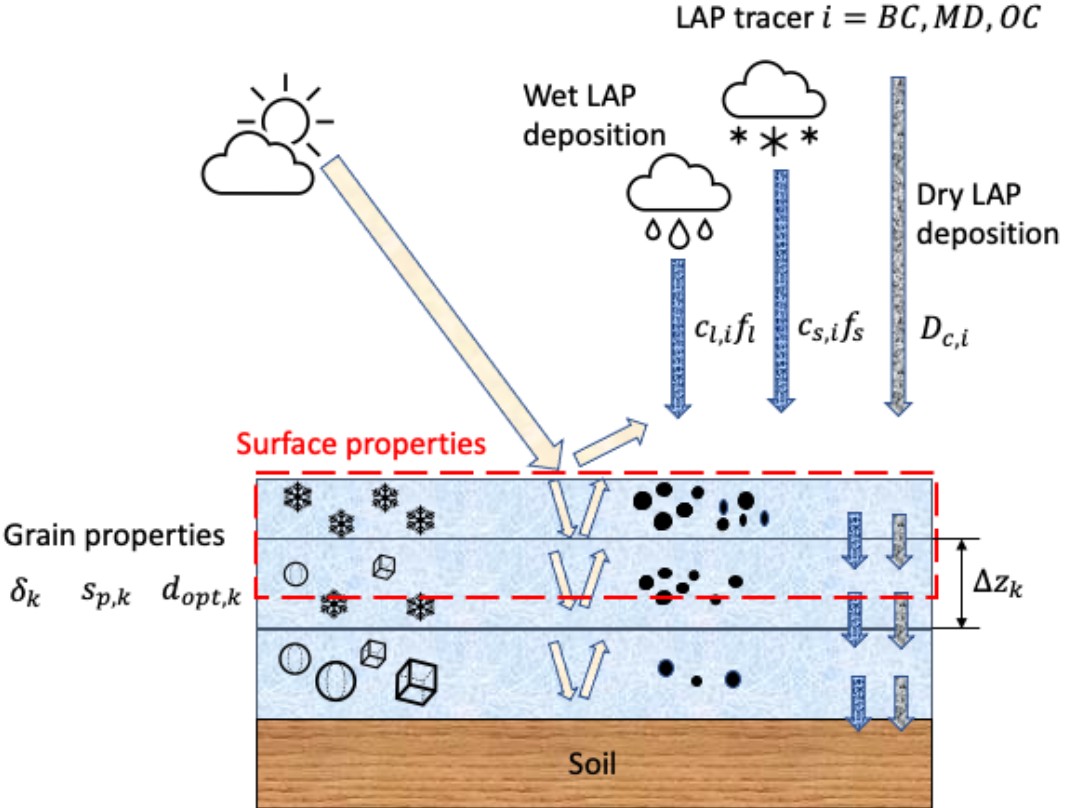

**Figure 1.** Conceptual scheme of the LAP deposition process implemented in GLASS and its effects on snow optical properties. The input of LAP mass in the snow is given by wet and dry deposition. Vertical exchange of LAPs withing the snowpack is driven by vertical water flow and is proportional to a specie-specific scavenging coefficient. The resulting concentration is near surface layer is used, together with surface snow properties, to predict snow albedo.

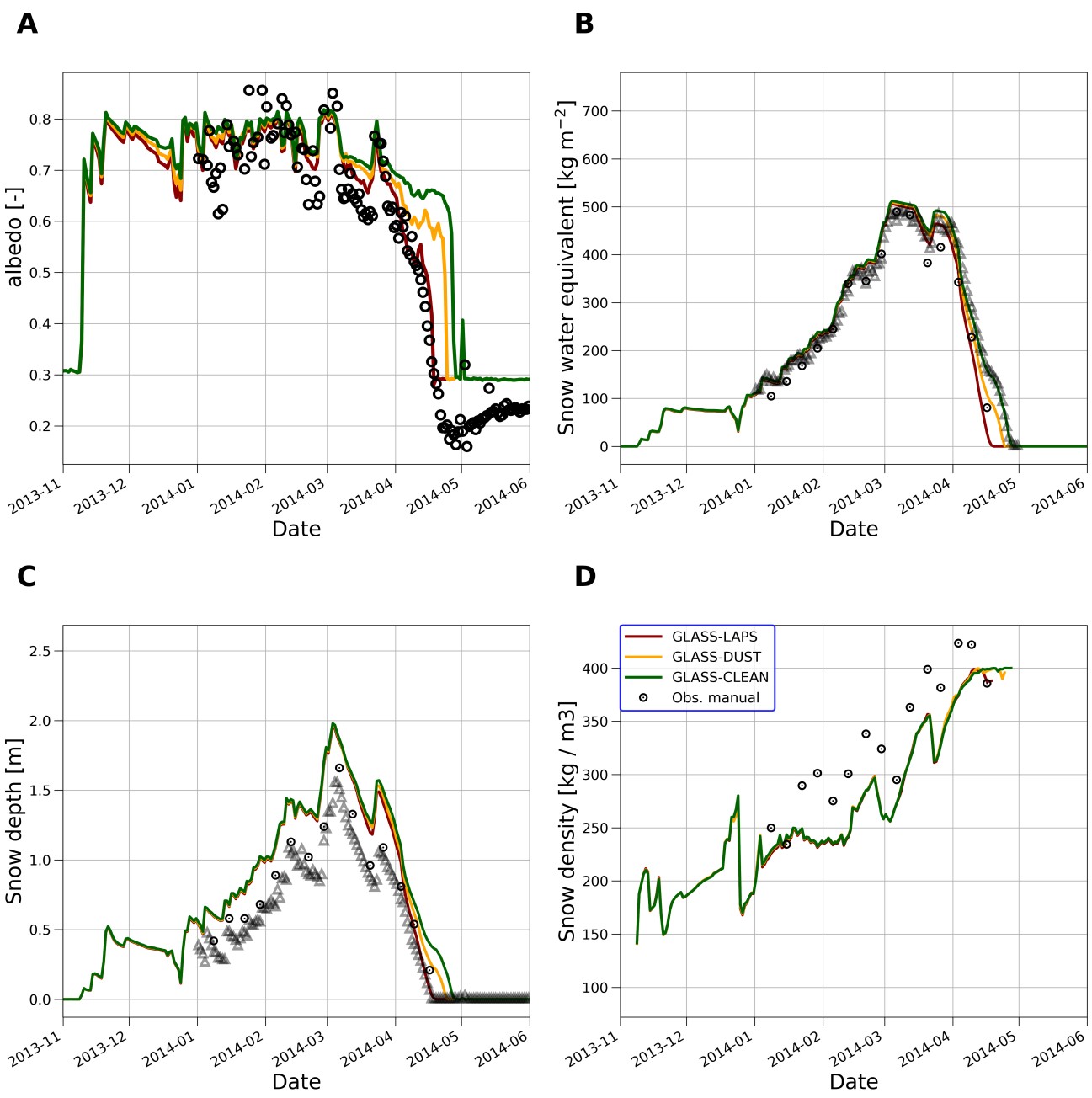

**Figure 2.** Results for the Col de Porte site during the 2013-2014 snow year. Simulated and observed values of daily surface albedo (panel A), SWE (panel B), snow depth (panel C) and snow density (panel D). Observations are reported as black markers (circles for manual obs., triangles for automatic obs.). Model simulations are reported for model forced by all LAPs (GLASS-LAPS, red lines), model forced by dust only (GLASS-DUST, tan lines) and by model with no impurities in snow (GLASS-CLEAN, green lines).

In order to compare modelled and observed surface albedo, modelled upward and downward shortwave radiation fluxes are averaged daily to correspond to observations.

For the *cdp* test site we find that GLASS reproduces daily snow albedo very well (Figure 2A) . We compare three model configurations, forcing the model with all three LAP species (GLASS-LAPS), forcing the model with mineral dust only (GLASS-DUST) and with no impurities deposited on snow (GLASS-CLEAN). The analysis reveal that i) the effect of dust and carbonaceous aerosols on albedo are comparable in magnitude for this site, and ii) the best match to daily albedo observation is provided by the GLASS-LAPS model configuration. Note that the model overestimates the albedo of underlying soil substrate. The reason for this mismatch is that the value of soil reflectivity was not calibrated for this particular location, and there is no vegetation in this configuration of the model. When considering snow water equivalent predictions (Figure 2B), the model matches observations very closely. During the ablation phase, some difference can be observed between model configurations, with snow disappearing about 10 days later in the case of clean snow compared to the configuration forced by all impurities. However, there are also comparable differences between automatic and manual snow depth observations in this period, with manual observations being closest to the GLASS-LAPS model prediction.

All three model configurations tend to overestimate snow depth throughout the winter with minimal differences between LAP treatments (Figure 2C). During the melt season, similarly to the results obtained for SWE, the closest match to observations is given by the models forced by impurities. The discrepancy between the excellent fit for SWE and the less good snow depth results can be explained by limitations in how snow density is represented in GLASS. However, the evolution of snow density simulated throughout the season exhibit a constant underestimation (Figure 2D), suggesting perhaps a discrepancy in initial density assigned to fresh snow. We note that because GLASS is designed for global scale climate predictions, these values are not calibrated for specific locations.

For the *cdp* site, spectral measurements were collected during 2014 and used to estimate snow SSA and impurities content (Dumont et al., 2017). The estimates of SSA vary significantly during the cold season in the range from 60 to 5 $\mathrm{m^2kg^{-1}}$. The snow model overall captures the magnitude and range of variations of SSA during the winter (Figure 3A). The decrease of SSA during the ablation season is qualitatively reproduced by the model, although observed values then to be somewhat lower than model predictions.

The concentration of LAPs in the near surface snow layer is also well captured by the model (Figure 3B), considering that forcing values are obtained from an atmospheric model climatology dataset. For this reason, we do not expect the model to closely match variations in $c_{eq,ns}$ throughout the entire snow season. The order of magnitude of observed LAP concentration is comparable with simulations, but intra-seasonal variations are not captured by the model. This could partially be due to the fact that the mixing ratio of LAPs in precipitation is assumed constant for each month. During spring, the snow ablation phase is characterized by a sharp increase in LAP concentration driven by the combined effect of sublimation and melt. During this phase, the increase in $c_{eq,ns}$ is overall well represented in the model.

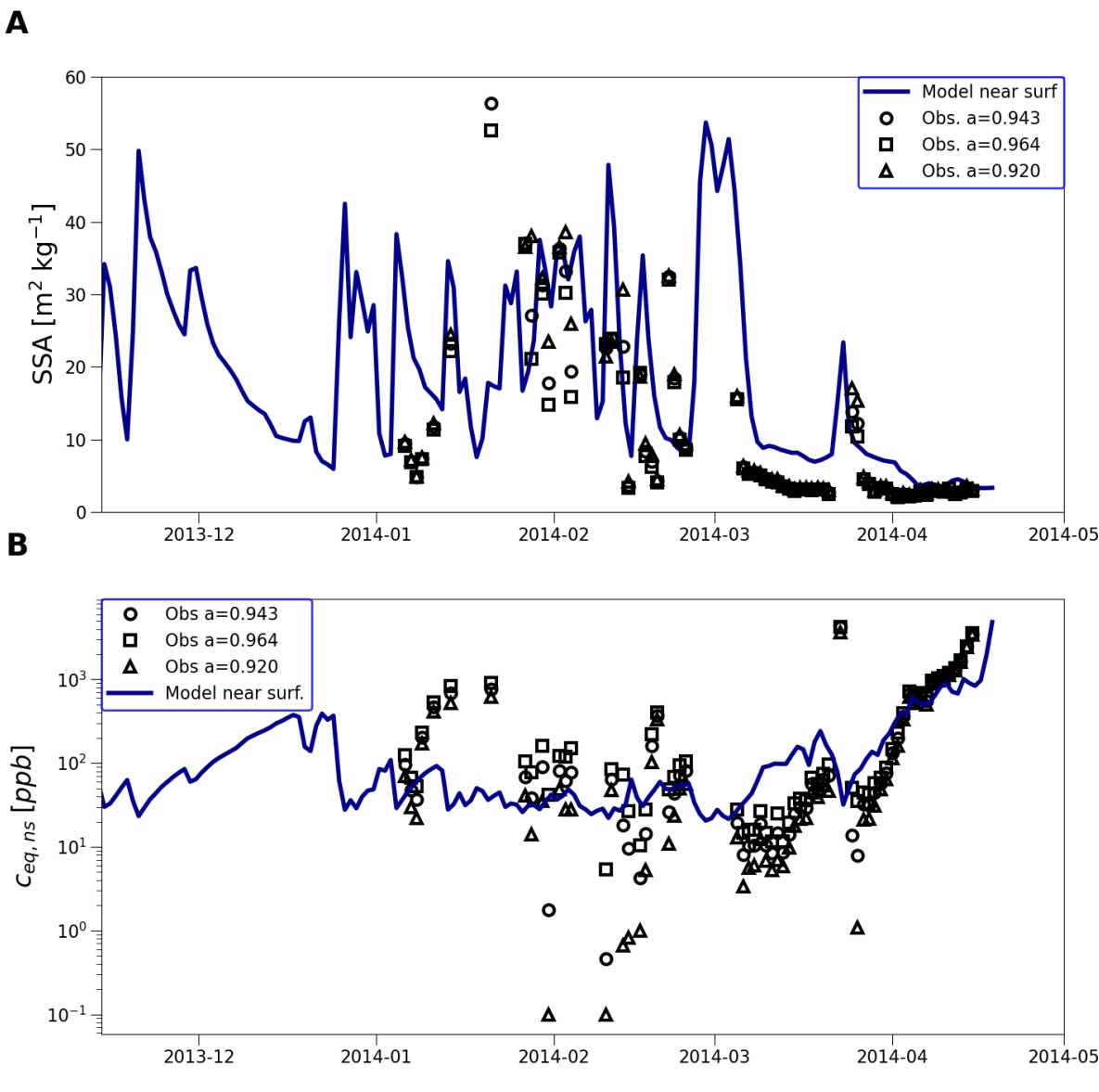

**Figure 3.** Results for snow specific surface area (panel A) and near-surface concentration of impurities (panel B) for the Col de Porte site (2013-2014 snow year). Spectral measurements carried out by (Dumont et al., 2017) are reported as black marker for different value of retrieval parameter $a$. Model simulations by GLASS-LAPS are reported as blue lines.

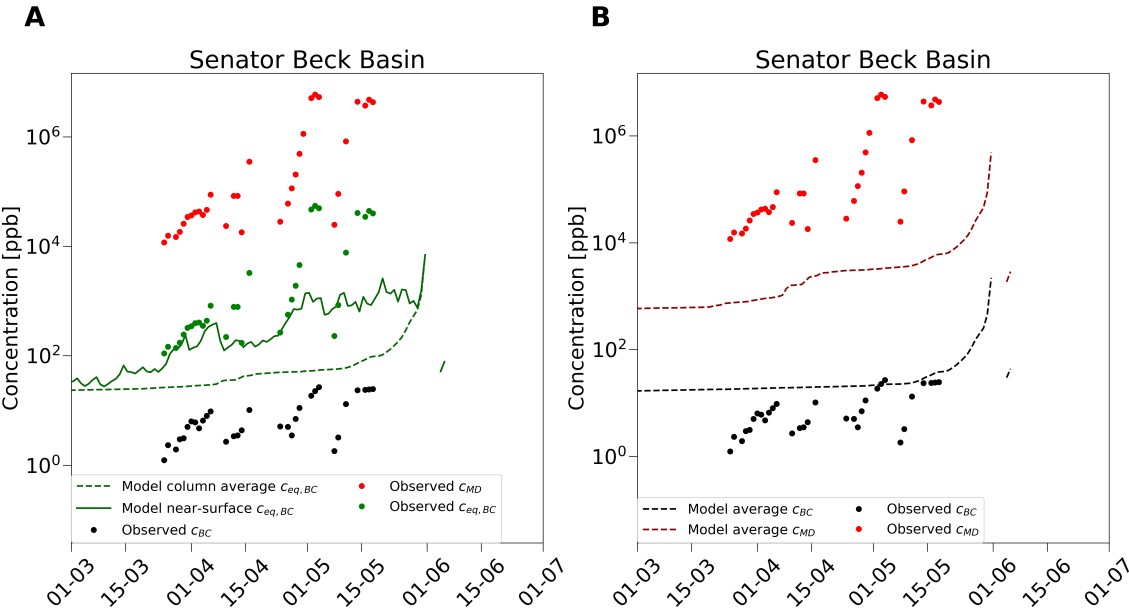

**Figure 4.** Comparison between modelled and observed LAP concentrations for the Senator Beck Basin (*snb*) site for spring 2013. Panel (A): Observed MD (red circles) and BC concentration (black circles) in the top 30 cm of snow. The total LAP concentration $c_{eq,BC}$ observed (greed circles) and modelled in the near-surface layer (green line) and averaged over the entire modelled snowpack (green dashed line) are also shown for comparison. Panel (B): The same observed MD (red circles) and BC concentration (black circles) are now compared to the vertically-averaged modelled concentrations of BC (black dashed line) and MD (red dashed line).

## 4.2 Predictions for Senator Beck Basin

We further compared modelled and observed concentration of impurities in snow by using a dataset collected from field campaigns at the Senator Beck Basin (*snb*) and Swamp Angel (*swa*) sites in Colorado (Skiles and Painter, 2015, 2017).

Figure 4 shows a comparison between observed and modelled concentration of LAPs in snow for the Senator Beck Basin site for the year 2013, which includes the occurrence of intense dust deposition events during spring. In Figure 4A we compare the total LAP content in snow as equivalent concentration of black carbon ($c_{eq,BC}$) as defined in eq. (10). During the first part of the year, the magnitudes of measured and modelled concentration over the model near-surface layer are comparable. However, during spring the intense LAP deposition events recorded at the site are underestimated by the model. We also note that the rapid increase in modelled LAP concentration at the end of the snow season happens later compared to the observations due a slower snow ablation. Average $c_{eq,BC}$ within the entire snowpack is lower than that modelled for the near-surface layer during the entire season, until the very end of the season. Figure 4B further compares MD and BC observations with the column-average model predictions. For dust, model predictions are lower than observations throughout the season, and again increase rapidly towards the end of the snow season driven by snow ablation. BC modelled concentrations are small throughout the season, and tend to be in better agreement with observed values (Figure 4B).

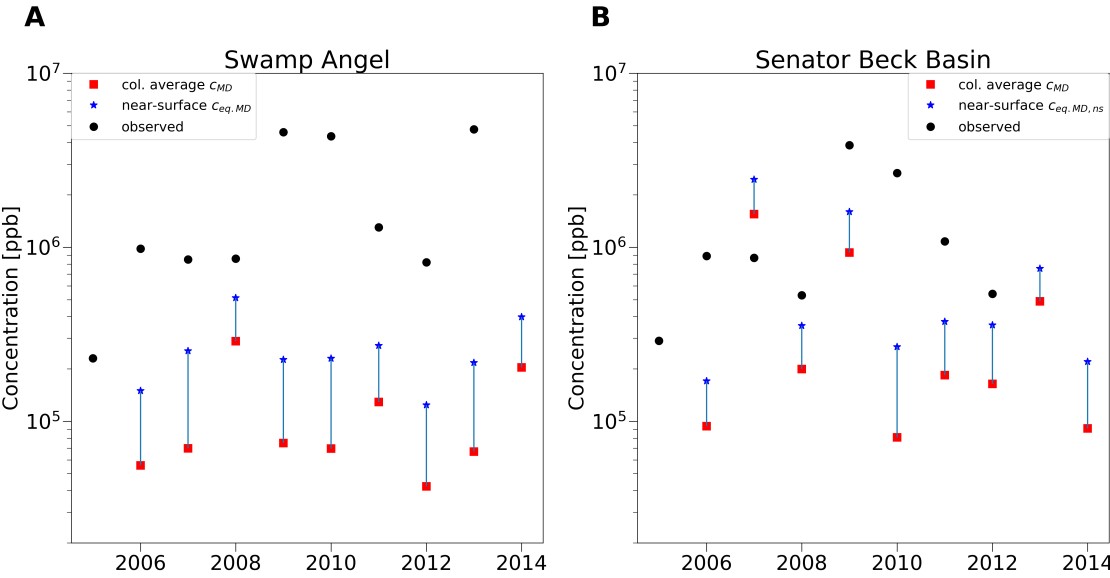

**Figure 5.** Comparison between modelled and observed end-of-year dust concentrations for the Swamp Angel (*snb*, panel A) and the Senator Beck Basin (*snb*, panel B) sites in southern Colorado. Dates are in format day-month. Data are obtained from Skiles and Painter (2015) and correspond to the average dust concentration $C_{MD}$ averaged over the top 30 cm of the snowpack (black circles). Model values are the column vertical average MD concentration $c_{MD}$ within the snowpack (red squares). The equivalent concentration of LAPs in the near-surface layer of the snowpack is also reported as comparison (blue star markers).

We extend this comparison by examining a multi-year dataset of dust concentration collected at the end of the snow season at the Senator Beck Basin (*snb*) and Swamp Angel (*swa*) sites, respectively an high elevation alpine, and a lower evelation "sub-alpine" site Skiles and Painter (2015). Note that again these observations correspond to average MD concentrations over the top 30 cm of the snowpack. As a comparison with observed data, we report both modelled MD concentration averaged over the snow column, and near-surface equivalent LAP concentration, expressed as dust content ($C_{eq,MD}$).

For the *swa* site, we find that the model underestimates observed concentrations throughout the season, with the near surface ($C_{eq,MD}$) being generally larger than the average snowpack concentration and closer to observations (Figure 5A).

For the *snb* site, the model still underestimates measured concentrations, but the underestimation is smaller than that observed for the *swa* site. Modelled concentration values at *snb* are larger compared to the case of *swa*, and exhibit a larger year-to-year variability (Figure 5B). Furthermore, the variability in observed concentrations between the two sites is significant, with the high-elevation site (*snb*) exhibiting lower dust concentration values, thus suggesting a large spatial heterotegeneity in dust content. In particular, the model underestimates dust concentration for the years characterized by extremely high dust loads at this site (in particular, year 2010).

## 4.3 Predictions for the SnowMIP sites

For some SnowMIP sites, daily albedo observations are available for model evaluation (Figure 6). During the accumulation phase, some underestimation of daily albedo by the model can be observed at some of the sites (*snb*, *sap*, and *wfj*). It is worth noting that some of the sites where the model exhibits the largest SWE underestimations correspond to sites where a negative bias in snow surface albedo is also reported (e.g., at *wfj* and especially at *sap*, which is the site characterized by the largest SWE and albedo underestimation in our dataset) so that surface albedo appears the primary source of the SWE bias. Furthermore, the temporal variability of modelled albedo is generally smaller than the observed one. This can be the result of effects due to the snow surface grain properties, and can also depend on how the solar angle is included the albedo parameterization through the coefficient $\phi_b(\mu)$. During ablation phase, the results are dependent on the model's ability to correctly capture the timing of complete snow melt. However, in general the decrease of albedo in the spring appears better reproduced by the model configurations with impurities, except in the case of *wfj*.

SWE model predictions show a good fit to observations at most SnowMIP sites (Figure 7). However, for some stations overestimation (*rme, snb*) or underestimation of SWE (*swa, sap, wfj*) are observed for the specific snow year examined. We note that, for the sites which exhibit the largest SWE discrepancies between model output and observations, the effect of LAPs predicted by the model does not appear to be the primary reason for model biases. In particular, accounting for LAPs does not appreciably change the seasonal peak SWE, which the model underestimates by about 22% at two of the sites (*swa* and wfj) and by about 34% at *sap*. For the sites in Colorado (*swa* and *snb*) we find that dust dominates the LAP effect on snow melt when compared to carbonaceous aerosols (Figure 7, panels A and B). The effect of LAPs is smallest for the Japanese site (*sap*) and for the one in Finland (*sod*). For the sites located in the Alps (*wfj*, in Figure 7D, and the already examined Col de Porte, *cdp*, featured in Figure 2B), we find that i) the effect of carbonaceous particles (BC and OC) is larger than simulated for the North American sites, and has the same magnitude as the effect of dust. When the model is forced with dust only, the number of snow days lost decreases significantly (and is between half and two-thirds of that predicted in the all-LAPs scenario).

Similarly, the skill in snow depth predictions varies across sites (Figure 8). Good performance are observed for *swa*, *sap*, and *sod*, although in the latter case some overestimation is observed during the boreal winter. A considerable overestimation of snow is observed for *snb* and *rme* sites, while the melt occur earlier then observed in *wfj*, case in which clean snow model would have the best fit.

To quantify the effect of LAPs, we examine the number of snow days predicted by the different model configurations. The decrease in number of snow days for the stations is reported in Table 2 (average and standard deviation across the years of the experiment) for the DUST only and all-LAPs scenarios. The largest average number of snow days lost is observed for *swa*, *snb* (dust-driven), and *wfj*. However, in the latter case the CLEAN scenario is actually the closest to the observations.

Figure 9 shows the differences in yearly maximum SWE values between CLEAN, LAPS and DUST runs. Here we find that the effect of LAPs on yearly maximum SWE peak is very small for all sites, as one would expect as the effect of LAPs is more pronounced during the ablation rather than during the snow accumulation phase.

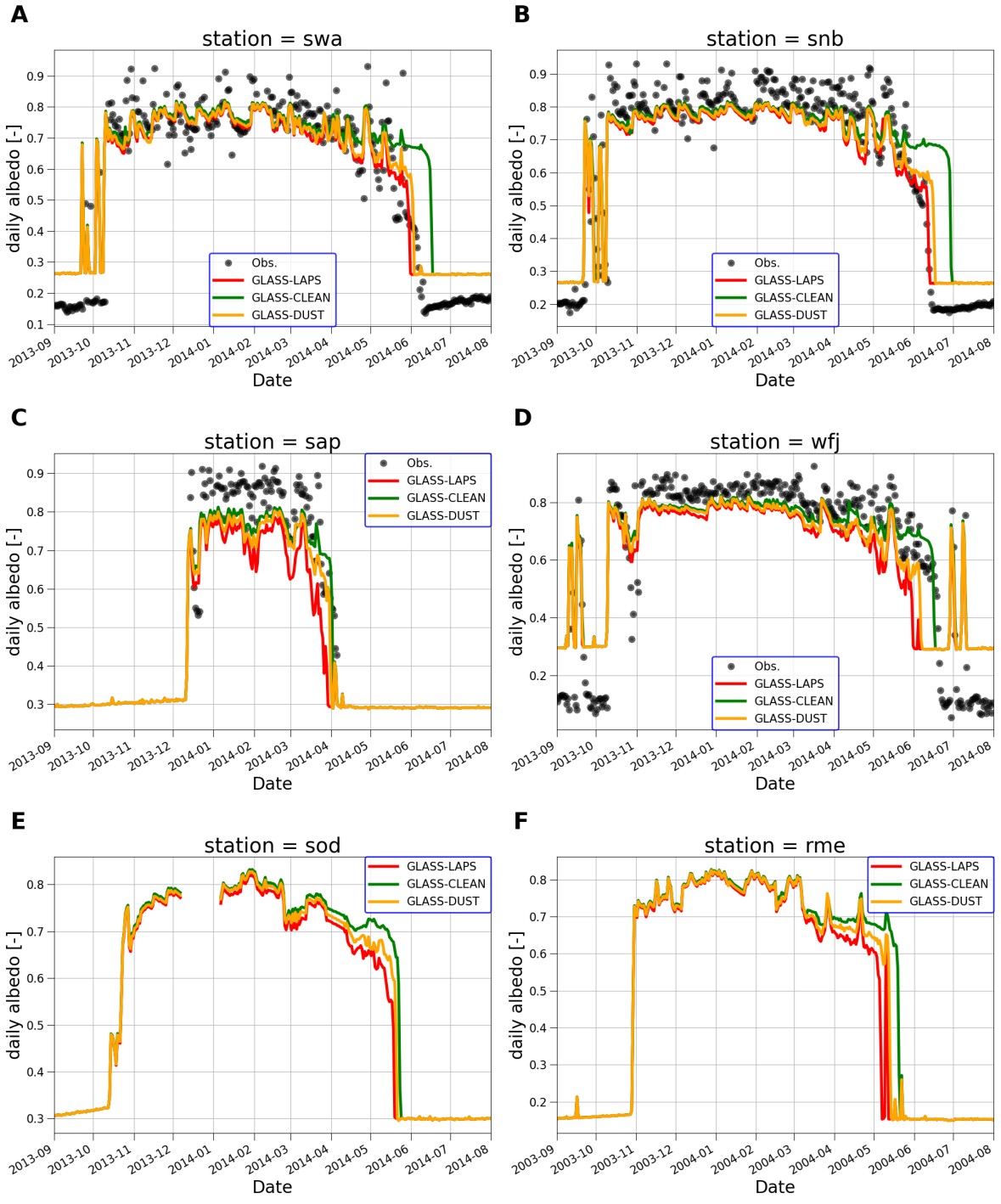

**Figure 6.** Daily average surface albedo predictions for 6 SnowMIP sites for a single snow year (snow year 2013-2014 for all sites except for *rme*). For each site, we compare model runs with all LAP species (GLASS-LAPS, red), with dust-only forcing (GLASS-DUST, tan), and with clean snow (GLASS-CLEAN, green). Where available, daily albedo observations are reported as reference (black circles).

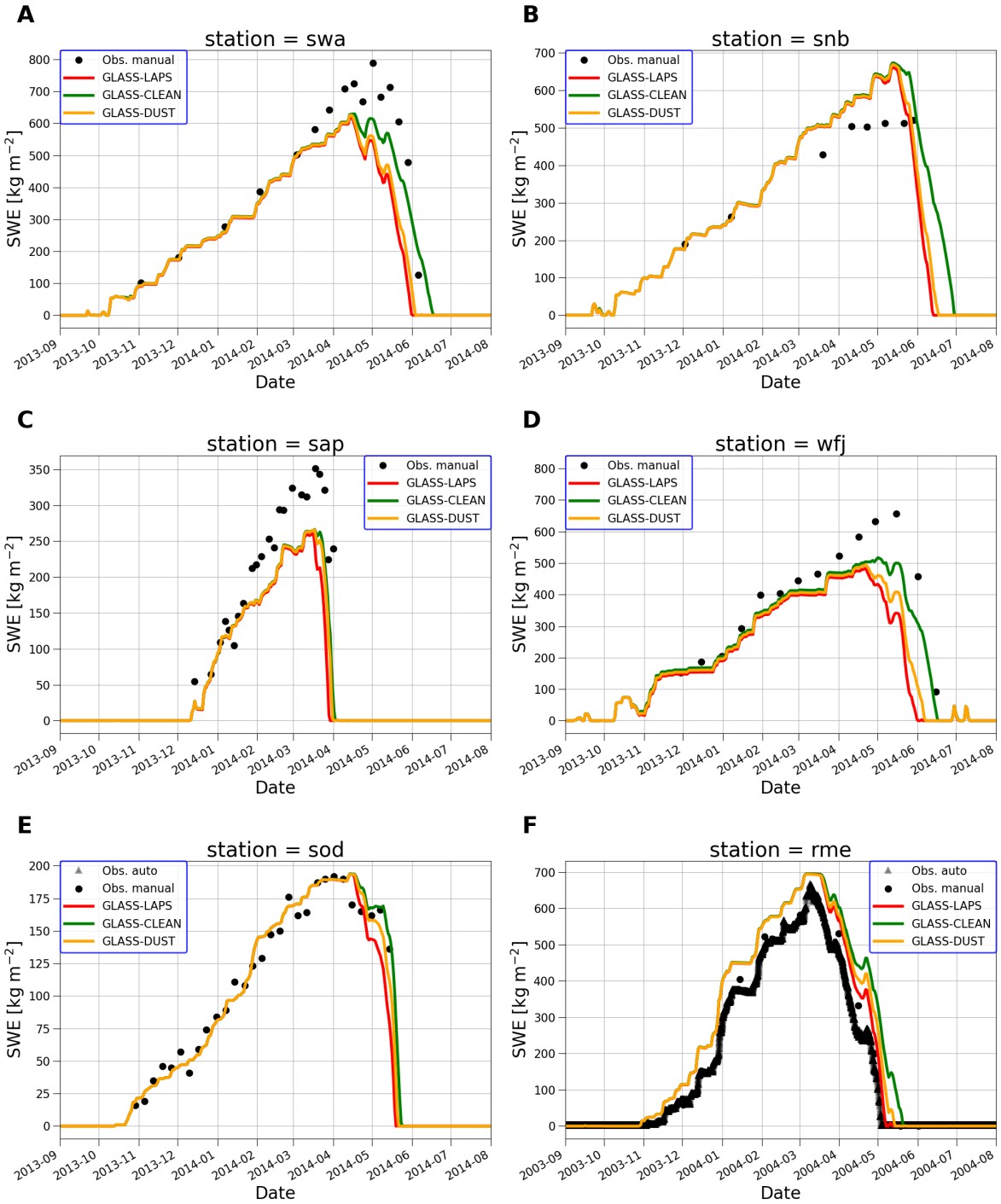

**Figure 7.** SWE predictions for 6 SnowMIP sites for a single snow year (snow year 2013-2014 for all sites except for *rme*). For each site, we compare model runs with all LAP species (GLASS-LAPS, red), with dust-only forcing (GLASS-DUST, tan), and with clean snow (GLASS-CLEAN, green). Where available, SWE observations are reported as reference (black circles for manual observations, black triangles for automatic observations).

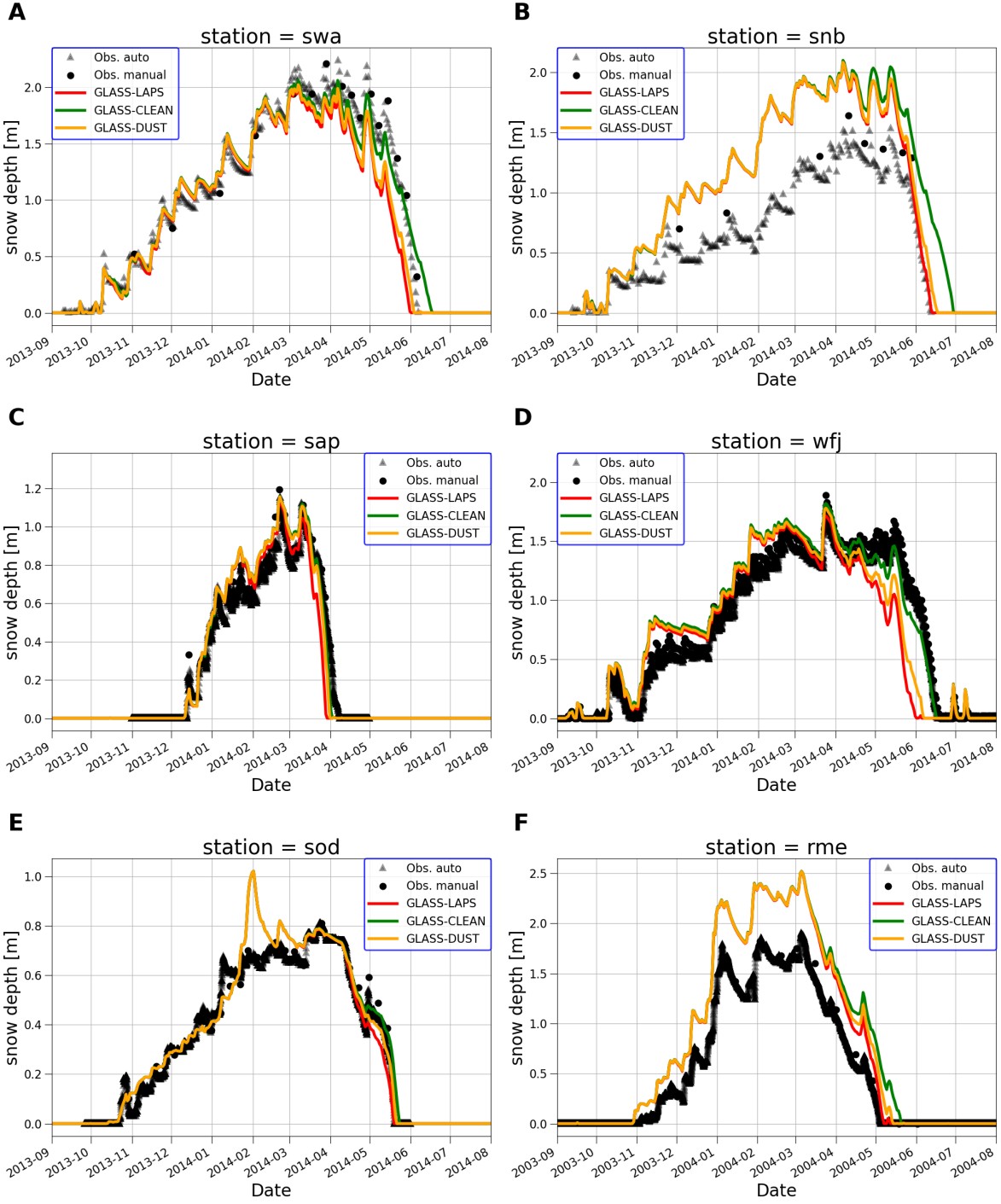

**Figure 8.** Snow depth predictions for 6 SnowMIP sites for a single snow year (snow year 2013-2014 for all sites except for *rme*). For each site, we compare model runs with all LAP species (GLASS-LAPS, red), with dust-only forcing (GLASS-DUST, tan), and with clean snow (GLASS-CLEAN, green). Where available, snow depth observations are reported as reference (black circles for manual observations, black triangles for automatic observations).

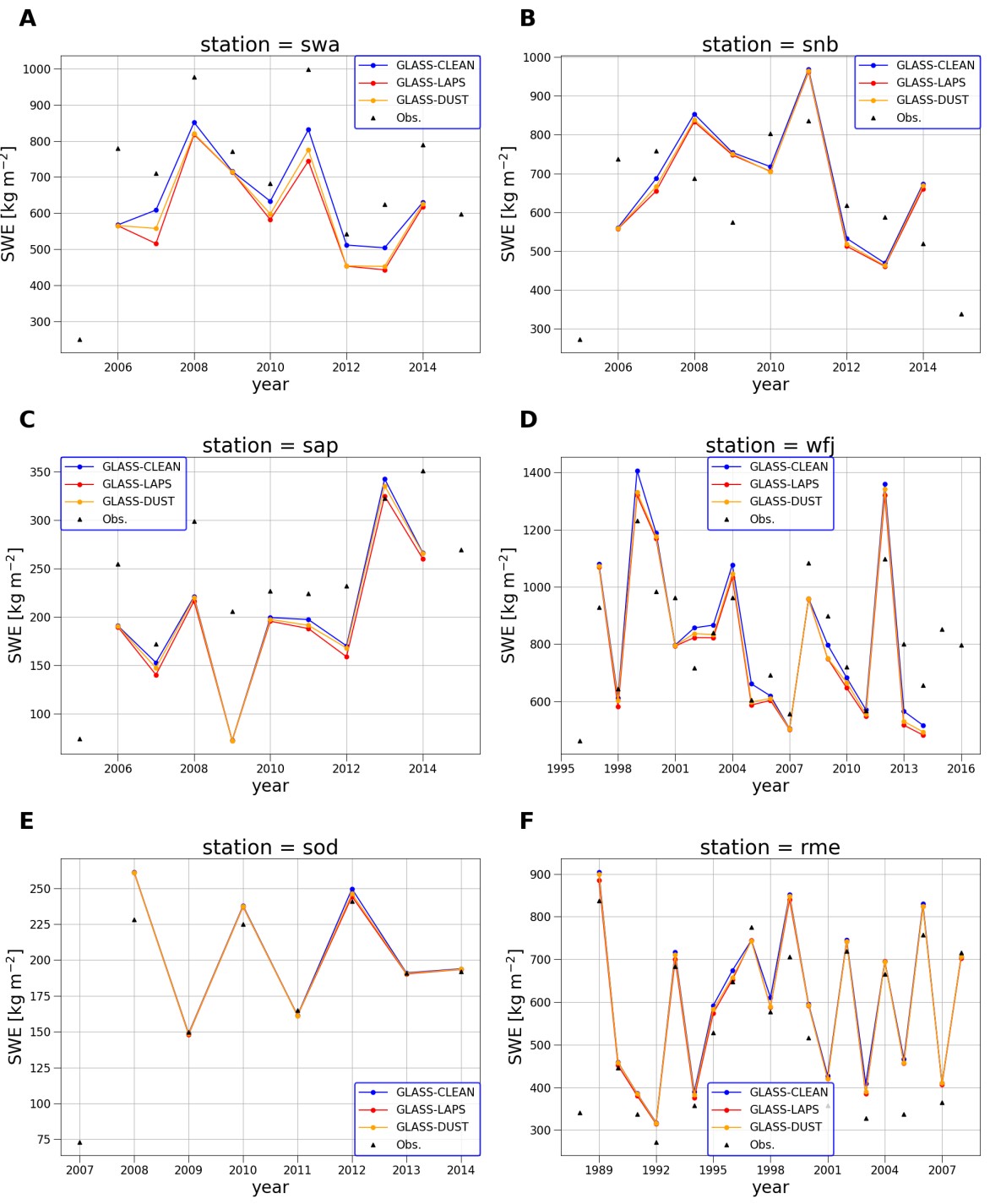

**Figure 9.** Snow water equivalent predictions for each year in different model configurations: With all LAP species (GLASS-LAPS, red), with dust-only forcing (GLASS-DUST, tan), and with clean snow (GLASS-CLEAN, blue).

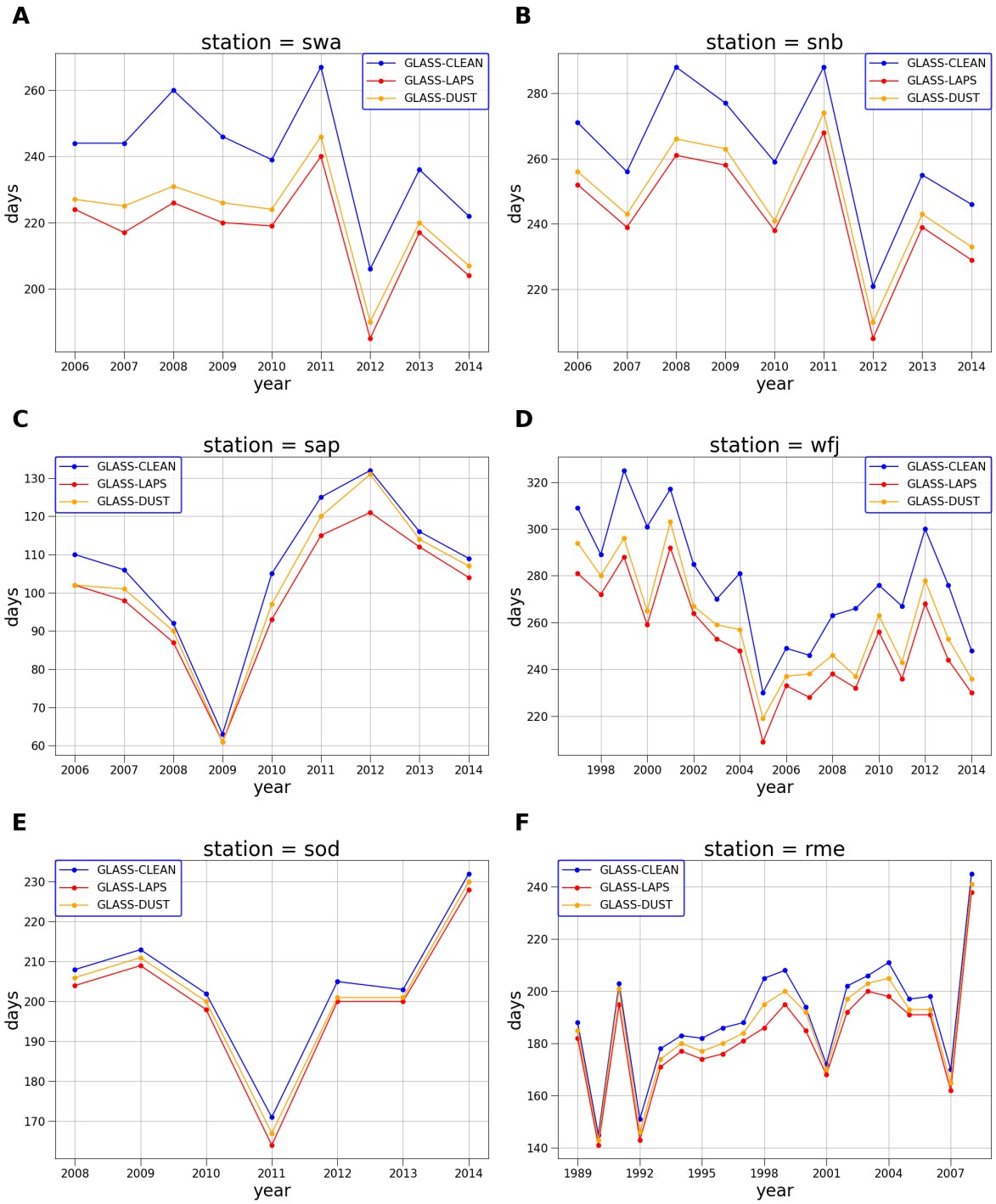

**Figure 10.** Number of snow days simulated for each year in different model configurations: With all LAP species (GLASS-LAPS, red), with dust-only forcing (GLASS-DUST, tan), and with clean snow (GLASS-CLEAN, blue).

|              |            | swa  | snb  | sap  | wfj  | sod | rme | cdp |
|--------------|------------|------|------|------|------|-----|-----|-----|
| CLEAN - LAPS | $\mu_t$    | 23.6 | 19.1 | 7.2  | 25.9 | 4.4 | 8.3 | 9.8 |
|              | $\sigma_t$ | 5.0  | 3.2  | 3.2  | 7.7  | 1.2 | 3.4 | 4.5 |
| CLEAN - DUST | $\mu_t$    | 18.7 | 14.7 | 3.9  | 18.2 | 2.6 | 4.4 | 5.7 |
|              | $\sigma_t$ | 4.2  | 3.2  | 2.6  | 7.7  | 0.9 | 2.0 | 3.2 |

**Table 2.** Decrease in the number of snow days in the LAPS and DUST cases compared to CLEAN, including temporal mean ($\mu_t$) and standard deviations ($\sigma_t$) of yearly values computed over the entire experiment period.

Figure 10 shows the differences in number of snow days values between CLEAN, LAPS and DUST runs. this number is quite significant for several of the sites. However, we find that the year-to-year variations in number of snow days lost to LAPs are generally contained.

## 5 Discussion

We found that the effect of dust is dominant over that of carbonaceous aerosols for the sites in North America (i.e., for the sites *snb*, *swa*, *rme*), where the combined effect of LAPs determine a significantly shorter snow season than would be in the case of clean snow (shorter by 24 and 19 days for the *swa* and *snb* sites, averaged over the entire time period of the observational records). In the case of the Senator Beck Basin site (*snb*), effects of similar magnitude have been reported by (Skiles and Painter, 2019), which found a dust-driven reduction of about 30 snow days, with previous estimates ranging between about 20 to 50 days depending on the LAP concentration in the spring (Skiles et al., 2012). Similarly, for the Swamp Angel site, in our analysis the melt date advances on average by 24 days, corresponding to the lower end of a previous study (Skiles et al., 2015). Our results on average indicate an effect of all LAPs on melt date which is the lower end of the range computed in these previous studies. While dust is less absorptive than BC, it is by far the most abundant by mass and at these sites appears to dominate the snow surface darkening. Similar results underlying the primary role played by mineral dust on snow melt have been reported for other regions. For example, Sterle et al. (2013) reported a similar behavior for snow in the Sierra Nevada, with mineral dust exhibit a larger impact of snow compared to black carbon. Over high-mountain Asia, Sarangi et al. (2020) similarly observed dust playing a fundamental role in the darkening of high-altitude snow.

On the other hand, we found that for the Alpine sites the effects of dust and carbonaceous particles are similar in magnitude. In the case of the Col de Porte site (*cdp*), the result we obtained (10 snow days lost due to all LAPs, on average) is slightly larger than the results from a previous modelling study (Tuzet et al., 2017), which reported snow melt date advancing 6 to 9 days in the 2013-2014 season depending on LAP parameters used in the analysis. For the *cdp* site, we were able to compare modelled snow near-surface properties (snow specific surface area and concentration of BC-equivalent concentration of LAPs) with in-situ spectral measurements for one snow season. This analysis showed that significant scatter exist between modelled and observed values. However, the order of magnitude of LAP concentration was captured by the model, as well as the increase in LAPs during the ablation season. We note here that since the snow scheme is forced by modelled deposition fluxes, a perfect

match to observation was not necessarily expected. Similarly, the increase in snow optical diameter (i.e., decrease in SSA) during the snow ablation phase was also captured by the model, although with some overestimation of observed SSA values.

## 5.1    Sources of uncertainty and model limitations

The modelled LAP concentration directly depends on how LAP scavenging processes in the snowpack are represented in the snow model. In the current GLASS formulation, scavenging coefficients are constant across snow layers, and the three

LAP species considered here do not account for the different behavior between e.g., hydrophobic and hydrophylic carbon components. Furthermore, scavenging coefficients do not depend on the mixing state of impurities (i.e., internally or externally mixed). Future extensions of the model could include more realistic scavenging parameterizations depending on the LAP mixing state.

     Uncertainty in modelled LAP concentrations and snow optical properties also depends on the dataset of input LAP deposition

fluxes used here. While the other atmospheric variables used to force the land model consists of in-situ observations, LAP deposition data are obtained from a reanalysis dataset. Therefore, LAP fluxes used here are coarser in space and time and may not be fully representative of the local deposition flux at the sites. Furthermore, here we assume that dust absorption properties are constant globally, while in general they do depend on dust mineralogy, which is spatially heterogeneous. The comparison with LAP concentration observations at two of the sites helped us constrain these sources of uncertainty and showed that

while modelled LAP concentrations exhibit discrepancies with observations, overall the order of magnitude and seasonal trend throughout a snow season are reproduced by the model.

     Furthermore, the model could be improved by using more detailed optical models such as the Two-stream Analytical Radiative TransfEr in Snow (TARTES) model (Libois et al., 2013) or the Snow, Ice, and Aerosol Radiative (SNICAR) model (Flanner and Zender, 2005), which are not limited to the case of semi-infinite snowpack as is the case for the albedo parameterization

used here. We found that both when considering clean snow and in the model configuration with LAPs, biases in modelled albedo are observed at some of the sites. Using a physically based model such as TARTES or SNICAR might help reduce these discrepancies. We note that despite the bias observed for specific sites, the model was developed for global applications and was not tailored to terrain or climate conditions of these sites. Thus, multiple physical processes may be at the origin of these discrepancies, as also discussed by Zorzetto et al. (2024), and should be the subject of future model development and testing

efforts. Furthermore, the current application was limited to sites with little to no vegetation. We believe further investigation of model performance and of the effects of LAPs for snowpacks in forested areas would be an important addition to this line of research.

     While this analysis focused on a point-scale application of the model, future extensions of the work should explore the ability of GLASS-LAPS to reproduce spatial statistics of snow cover using sub-grid tiling schemes and a description of topographic

effects (e.g., Chaney et al. (2018); Zorzetto et al. (2022)). While the model was here tested in a land-only configuration forced by an offline atmosphere, LM4.1 and GLASS are designed as components of an Earth System Model and are thus tailored to coupled simulations with an atmospheric model. In this coupled model configuration, currently under development, the computation of the monthly constant mixing ratio of LAPs in precipitation will no longer be needed as both liquid/frozen

water fluxes and LAP deposition fluxes will be provided by the same atmospheric model. This would further reduce one of the
sources of uncertainty here, due to the coarse temporal resolution of LAP deposition fluxes.

## 6 Conclusions

In this work we extended a recently developed numerical snow model (GLASS) to include the physical processes connected with the deposition of LAPs. GLASS, implemented in the GFDL land model, can be used in global scale, centuries-long climate simulations. We tested the new model configuration over a set of SnowMIP sites, forcing the model with in-situ meteorological data and with LAP deposition rates obtained from a general circulation model (AM4.0). We found that the model satisfactorily represents seasonal snow amounts over the test sites, although performance is to a certain extent location dependent. This is not surprising given the large variability in climates, as well as surface properties and terrain types. Running the model with clean snow, dust only and forced by all LAPs allowed us to investigate the relative contribution of different aerosol species to snow melt. We found that the effect of LAPs on snowpack evolution is significant at all sites examined, with an average number of snow days lost due to LAPs varying between 5 and 24. For sites in the Western US, the effect of dust is predominant and is responsible for most of the LAP-driven melt. In other locations this is not the case: In the sites in the Alps, for instance, carbonaceous aerosols play a larger role relative to dust. Our results support large-scale applications of the new model configuration to simulate snowpack globally, under historical and projected climate condition. These analysis will be the objective of future work.

*Code and data availability.* The code and data used in this study are freely available online at https://zenodo.org/records/10901373. The comparison with observed LAP data is available at https://zenodo.org/records/14043054

*Author contributions.* All authors contributed to research design and to writing the manuscript. E.Z. Developed software, performed analysis of model results, and drafted a first version of the manuscript.

*Competing interests.* The authors declare no competing interest.

*Disclaimer.* TEXT

*Acknowledgements.* The authors acknowledge funding from the NOAA Climate Program Office (CPO) grant number NA18OAR4320123 "3D-Land Energy and Moisture Exchanges: Harnessing High Resolution Terrestrial Information to Refine Atmosphere-to-Land interactions

in Earth System Models". The authors thank Dr. Justin Perket and Dr. Nicole Schlegel at NOAA GFDL for reviewing a first draft of the manuscript.

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
