# Peer review of "Quantifying radiative effects of light-absorbing particles deposition on snow at the SnowMIP sites"

_EGUsphere, 2024_

## Author Comment (AC1)

Reviewer #2

The authors expanded the GFDL LM4.1 snowpack processes to include LAPs effects and evaluate the simulated snow albedo across different measurement sites. They found that the resulting snow model with LAP-aware snow reflectivity show a good agreement with measurements of broadband albedo and seasonal SWE over the study sites. They further evaluated the number of snow-days lost due to the deposition of dust and carbonaceous aerosols, which ranges between 5 and 24 days depending on locations. This work is an important improvement for the GFDL snowpack model, which could provide better simulations for future coupled climate runs. Overall, the manuscript is well organized. I have a few specific comments/suggestions for the authors to consider.

We thank the reviewer for providing insightful comments on our work. Please find below a point-by-point response to the comments on the manuscript. We believe addressing these comments would significantly improve the manuscript with respect to our first submission. We report the review in black, and our response in light blue. Excerpts from the revised manuscript with the proposed changes are reported here in *Italic*. The line numbers referenced in our response refer to the position of the excerpts in the proposed revised manuscript.

Specific comments:

1. Is there any canopy snow process included in LM4.1? A brief description would be useful.

   The land model GFDL LM4.1 in general resolves snow intercepted by canopy layers, computing its full mass and energy balance. The albedo of leaves is a weighted average which considers the amount of fractional leaf area covered by snow. However, in the current application vegetation is not considered for the sites. In this paper, we limited our analysis to experimental sites with little to no vegetation cover, in order to focus our attention on the effect of impurities rather than on the complex snow-vegetation interactions. Therefore, we have turned off vegetation in the model, which effectively is not allowed to grow throughout the simulation. We have expanded the discussion on this model setting in the manuscript, and we now note that further investigations focusing on forested sites would be an important topic of future research. We propose to add the following text to the manuscript (at line 126 of the revised manuscript):

   *"In LM4.1, vegetation is represented by a set of plant cohorts which evolve dynamically. Multi-layer vegetation canopies interact with the surface via multiple processes, including turbulent exchange of mass and energy, the transfer of longwave and shortwave radiation, and the interception of liquid and frozen precipitation. For additional details, the reader is referred to*

*Shevliakova et al., (2024). In this application, we have decided to limit our analysis to sites with little to no vegetation in order to focus our attention on snow and LAP deposition processes. Therefore, vegetation is turned off and canopy layers are not present in the model simulations, following the experimental setup used in Zorzetto et al. (2024)."*

However, we believe that further validation of the model proposed here over forested areas would be an important and informative future research objective. We proposed to discuss this in the discussion section of the revised manuscript at line 475:

*"Furthermore, the current application was limited to sites with little to no vegetation. We believe further investigation of model performance and of the effects of LAPs for snowpacks in forested areas would be an important addition to the current research."*

2. How is the snowpack liquid water treated? Is there a liquid water holding capacity parameter prescribed? How important is it to assign different inter-layer snowmelt water scavenging coefficients for IM vs EM LAPs?

We agree that the treatment of liquid water flow was not discussed in the original paper, and it is a relevant process for the transport of LAPs. We propose to include a detailed description of these processes instead of just referring the reader to the model description paper (Zorzetto et al., 2024). We propose to include the following discussion at line 162:

*"At each model time step, the full energy and water mass balance of the snowpack is solved using an implicit numerical formulation, which is required for land-atmosphere coupled model runs with relatively coarse time step (30 min) for which the GLASS was designed. After performing the vertical heat balance of the snowpack, the temperature change and change of phase is evaluated for all snow layers. The vertical balance of liquid water is then evaluated throughout the snowpack, with liquid precipitation providing the upper boundary condition. The snow density is used to evaluate the pore space available for liquid water in each snow layer. Following Vionnet et al., (2012), the liquid water holding capacity for a snow layer k with thickness $\Delta z_k$ and local solid-phase density $\rho_{s,k}$ is given by*

$$W_{liq,max,k} = 0.05 \rho_w \Delta z_k \left( 1 - \frac{\rho_{s,k}}{\rho_i} \right)$$

*with $\rho_i$ the density of ice, and $\rho_w$ that of liquid water. For a more detailed description, see Zorzetto et al., (2024)."*

With respect to the second point, we now clarify that we are using a single scavenging coefficient for IM and EM LAPs, and propose to discuss potential extensions of the model with

more realistic scavenging parameterizations. We propose the following addition at line 453 of the revised manuscript:

*"The modelled LAP concentration directly depends on how scavenging processes of LAPs in the snowpack are modelled. In the current model formulation, scavenging coefficients are constant across snow layers, and the three LAPs species considered here do not consider different behavior between e.g., hydrophobic and hydrophylic carbon components. Furthermore, scavenging coefficients do not depend on the mixing ratio of impurities (i.e., internally or externally mixed). Future extension of the model could include more realistic scavenging parameterization depending on the LAP mixing state."*

3.  How did the authors assign dry and wet deposited LAPs to IM or EM within the snow?

Dry deposition contributes to EM impurities in the snowpack, while wet deposition (from both liquid and frozen precipitation) contributes to IM LAPs. We propose to clarify this feature of the model in the revised manuscript. The proposed addition at line 181 would read:

*"The mass of LAPs added to the snowpack by dry deposition is assumed to be externally mixed (EM), while LAPs deposited as wet deposition either due to liquid or frozen precipitation contribute to IM LAPs within the snowpack. In the model, the state of mixing of LAPs within the snow (IM or EM) does not change as a consequence of melt and freeze cycles occurring in a snow layer."*

4.  How many dust size bins are considered and what are they?

We have considered a single bin size with a single absorption value for dust. This is clearly a limitation, but can be extended in future studies provided that dust bins sizes and their respective optical properties are available as model input data. We now propose to explicitly mention this in the manuscript (line 176):

*"The model considers a single LAP size distribution for mineral dust, without tracking separately dust particles of different sizes. "*

5.  Does melt-freeze of snow change the IM or EM status of LAPs?

In its current configuration, melt/freeze cycles do not change the IM/EM status of LAPs within the snowpack. We proposed to clarify this point in the revised manuscript (line 181):

*"The mass of LAPs added to the snowpack by dry deposition is assumed to be externally mixed (EM), while LAPs deposited as wet deposition either due to liquid or frozen precipitation contribute to IM LAPs within the snowpack. In the model, the state of mixing of LAPs within the snow (IM or EM) does not change as a consequence of melt and freeze cycles occurring in a snow layer."*

6.  Is internal heating within the snowpack column due to light absorption considered?

Yes, as mentioned in the original manuscript we do consider the penetration of light within the snowpack and related heating rates. W propose to discuss this feature of the model in the revised manuscript as follows:

260  For thick enough snowpack (snow depth $> 0.02$ m), solar radiation penetrates within the snowpack and absorbed radiation is distributed exponentially

$$Q_s(z) = \sum_{b=1}^{2} (1 - \alpha_b) R_{s,b} e^{-\beta_b z} \tag{6}$$

where for each band $b$ $R_{s,b}$ is the downward shortwave radiation at the surface, and $\beta_b$ describes the penetration of light within the snowpack. The extinction coefficients for visible and near infrared light are estimated as in Jordan (1991) and Shrestha et al. (2010): $\beta_{NIR} = 400$, and $\beta_{VIS} = 0.003759 \, \rho \, d_{opt}^{-0.5}$, with density and optical diameter averaged over the near-surface layer of the snowpack, up to a maximum depth of 3 cm.

7. The snow albedo parameterization from Dang et al. 2015 and He et al. 2018 is for semi-infinite snowpack, which may lead to uncertainties in the albedo calculation here. It will be good to clarify this and briefly discuss this.

We agree that this limitation of the model should be discussed in the manuscript. The snow albedo parameterization employed is indeed for semi-infinite snowpack and thus can give rise to errors in the case of thin snowpacks. We propose to clarify this point in the revised manuscript, instead of just referring to Zorzetto et al. (2024).

Furthermore, in the revised manuscript we propose to explicitly describe how thin snowpacks are treated in the model. In particular, we mention that in the case of thin snowpacks, a snow area fraction is computed and used for albedo calculations. We propose to add the following to the model description:

265  Note that the albedo parameterization employed here based on the work of Dang et al. (2015) and He et al. (2018a) is derived for a semi-infinite snowpack, and thus in general can lead to biased albedo estimate for thin snowpack. In the case of this snowpack, the model computes a fractional snow cover $f_{snow}$ based on snow depth as follows

$$f_{snow} = \frac{h_s}{h_s + h_{s,c}} \tag{7}$$

with $h_s$ the snowpack depth, and $h_{s,c} = 0.0167$ m. In case of fractional snow cover, surface albedo is computed as a weighted
270  spatial average of snow and snow-free substrate optical properties.

In the discussion we propose to further discuss this issue by stating that (line 467):

*"Furthermore, the model could be improved by using more detailed optical models such as the Two-stream Analytical Radiative TransfEr in Snow (TARTES) model (Libois et al, 2013) or the Snow, Ice, and Aerosol Radiative (SNICAR) model (Flanner and Zender, 2005), which are not limited to the case of semi-infinite snowpack as is the case for the albedo parameterization used here. "*

8. How did the authors use the snow grain shape parameters to compute the optical diameter?

Prompted by this comment, we plan to expand the description of how snow grain size and shape are used to compute snow albedo. At line 249 of the revised manuscript, we propose to add the following:

*"The snow albedo parameterization used here explicitly accounts for the effects of snow grain size (through the snow grain effective radius) and shape on its optical properties. He et al., (2018) introduced Eq. (3) and provided the set of parameters $b_0, b_1$, and $b_2$ tabulated for four different snow grain shapes (sphere, spheroid, hexagon, and Koch snowflake). In GLASS, snow microphysics in each snow layer is parameterized by two parameters (snow sphericity and dendricity) which evolve in time due to the combined effect of dry and wet snow metamorphic processes, as well as due to wind effects (Zorzetto et al., 2024). The coefficients used in eq. (3) are selected at each time step based on snow shape properties in the near-surface snow layer: High-dendricity snow ($\delta_{ns} > 0.5$) is idealized as a collection of Koch snowflakes. Snow with lower dendricity parameter is considered as a collection of spheres (if sphericity parameter ($s_{p,ns} > 0.8$), spheroids (if $0.8 > s_{p,ns} > 0.2$), or hexagonal crystals (if $s_{p,ns} < 0.2$)."*

9. How is the alpha_b in equation (5) computed? What is the physical meaning of this parameter?

Here alpha_b is the snow surface albedo, which is computed according to eq. (2) based on physical properties on the snowpack (i.e., grain size and shape). We propose to clarify this by updating the notation used in the paper, and using the suffix b for snow surface albedo also in eq. (2), so as to clarify the dependence of albedo on the shortwave band (b = VIS or NIR for visible or near infrared band, respectively).

10. For daily average of albedo, did the authors use downward solar radiation as the weights?

Yes, we indeed used dimensional quantities as model output (downward and upward radiation fluxes) and computed their ratio at the daily time scale at which results are displayed. We now propose to explicitly mention this in the manuscript to clarify the procedure used in the analysis. At line 326 of the revised manuscript we propose to add the following:

*"In order to compare modelled and observed surface albedo, modelled upward and downward shortwave fluxes are averaged daily to correspond to observations"*

11. Figure 2A: Why does the model have a consistently high snow albedo than observations after 2014 May (which seems to be a snow-free period)? Also, since there is no snow after May 2014, why is the snow albedo from both observation and model not zero?

We now clarify that the quantity shown both in Figure 2A and Figure 5 of the original manuscript is the surface albedo, averaged at the daily time scale, which does not always correspond to the snow albedo. This is the same quantity provided in the observational datasets used for comparison. We propose to clarify this in the figure captions to explain why albedo values are shown even for snow-free periods. Furthermore, in the revised paper we propose to mention that for thin snowpacks the model computes an effective snow fractional area and therefore surface albedo is effectively based on a weighted spatial average of snow and snow-free substrate optical properties. At line 265 of the revised manuscript we propose to add the following:

265    Note that the albedo parameterization employed here based on the work of Dang et al. (2015) and He et al. (2018a) is
       derived for a semi-infinite snowpack, and thus in general can lead to biased albedo estimate for thin snowpack. In the case of
       this snowpack, the model computes a fractional snow cover $f_{snow}$ based on snow depth as follows

$$f_{snow} = \frac{h_s}{h_s + h_{s,c}} \qquad (7)$$

       with $h_s$ the snowpack depth, and $h_{s,c} = 0.0167$ m. In case of fractional snow cover, surface albedo is computed as a weighted
270    spatial average of snow and snow-free substrate optical properties.

12. Section 4: the description of the results is too qualitative. Please include some quantitative numbers when presenting the results. Also, more physical explanations/insights could be added to the results. For example, why does the model not capture the variation of the daily albedo variation over most sites (Figure 4) and why does the model results show systematic overestimate/underestimate in some sites (Figure 5).

We propose to add to the results section a more quantitative discussion of the discrepancies between modeled and observed snowpack properties as suggested in this comment.

At line 404 of the revised manuscript we propose to discuss the bias in SWE:

*We note that, for the sites which exhibit the largest SWE discrepancies between model output and observations, the effect of LAPs predicted by the model does not appear to be the primary reason for model biases. In particular, accounting for LAPs does not appreciably change the seasonal peak SWE, which the model underestimates by about 22% at two of the sites (swa and wfj) and by about 34% at sap.*

For albedo, at line 342:

*"During the accumulation phase, some underestimation of daily albedo by the model can be observed at some of the sites (snb, sap, and wfj). It is worth noting that some of the sites where the model exhibits the largest SWE underestimations correspond to sites where a negative bias in snow surface albedo is also reported (e.g., at wfj and especially at sap, which is the site*

*characterized by the largest SWE and albedo underestimation) so that surface albedo appears the primary source of the SWE bias.”*

And at line 397 we propose to add:

*Furthermore, the temporal variability of modelled albedo is generally smaller than the observed one. This can be the result of effects due to the snow surface grain properties, and can also depend on how the solar angle is included the albedo parameterization through the coefficient $\Phi_b(\mu)$.*

Now at line 459 we propose to add a comparison with observed concentration of LAPs:

*“Uncertainty in modelled LAP concentrations and snow optical properties also depends on the dataset of input LAP deposition fluxes used here. While the other atmospheric variables used to force the land model consists of in-situ observations, LAP deposition data are obtained from a reanalysis dataset. Therefore, LAP fluxes used here are coarser in space and time and may not be fully representative of the local deposition flux at the sites. The comparison with LAP concentration observations at two of the sites helped us constrain these sources of uncertainty and showed that while modelled LAP concentrations exhibit discrepancies with observations, overall the order of magnitude and seasonal trend throughout a snow season are reproduced by the model.”*

13. Figure 7: It seems that the model SWE bias is dominated by other model snow or forcing processes instead of LAP effects. This may be worth some discussion.

We agree and propose to discuss this at line 403 of the revised manuscript as follows:

*“SWE model predictions show a good fit to observations at most SnowMIP sites (Figure 7). However, for some stations overestimation (rme, snb) or underestimation of SWE (swa, sap, wfj) are observed for the specific snow year examined. We note that, for the sites which exhibit the largest SWE discrepancies between model output and observations, the effect of LAPs predicted by the model does not appear to be the primary reason for model biases. In particular, accounting for LAPs does not appreciably change the seasonal peak SWE, which the model underestimates by about 22% at two of the sites (swa and wfj) and by about 34% at sap.*

14. I would suggest adding a subsection for uncertainty discussion. Some of the uncertainties involved in the model are mentioned in my earlier comments. A few key uncertainty factors that are worth discussing: (1) snow grain shape and size evolution, (2) using Eq.8 to combine different LAPs, (3) aerosol deposition flux, (4) missing snowpack processes, (5) LAP meltwater scavenging, etc.

We agree. Given the number of processes involved we believe that a complete quantitative assessment of model uncertainty is beyond the scope of the present work. However, discussing these sources of uncertainty is important and helps to assess our results.

We propose to add a subsection to the Discussion section entitled *"Sources of uncertainty and model limitations"* where we will expand our discussion of model limitations and sources of uncertainty. At line 452 of the revised manuscript we propose the following discussion subsection:

*"*

*5.1 Sources of uncertainty and model limitations*

*The modelled LAP concentration directly depends on how scavenging processes of LAPs in the snowpack are represented in the snow model. In the current GLASS formulation, scavenging coefficients are constant across snow layers, and the three LAPs species considered here do not consider different behavior between e.g., hydrophobic and hydrophylic components for carbonaceaous LAPs. Furthermore, scavenging coefficients do not depend on the mixing state of impurities (i.e., internally or externally mixed particles). Future extension of the model could include more realistic scavenging parameterization depending on the LAP mixing state.*

[revised manuscript text omitted]

Libois, Q., et al. "Influence of grain shape on light penetration in snow." *The Cryosphere* 7.6 (2013): 1803-1818.

Flanner, Mark G., and Charles S. Zender. "Snowpack radiative heating: Influence on Tibetan Plateau climate." *Geophysical research letters* 32.6 (2005).

Shevliakova, E., et al. "The land component LM4. 1 of the GFDL Earth System Model ESM4. 1: Model description and characteristics of land surface climate and carbon cycling in the historical simulation." *Journal of Advances in Modeling Earth Systems* 16.5 (2024): e2023MS003922.

Vionnet, Vincent, et al. "The detailed snowpack scheme Crocus and its implementation in SURFEX v7. 2." *Geoscientific model development* 5.3 (2012): 773-791.

Flanner, Mark G., et al. "Present‑day climate forcing and response from black carbon in snow." *Journal of Geophysical Research: Atmospheres* 112.D11 (2007).

---

## Author Response (AR1)

**Response to reviewers**

Please find below our point-by-point response to the reviewer's comments. We report referee comments in black, and our response and excerpts from the revised manuscript in blue. Line numbers referenced below refer to the revised manuscript unless otherwise noted.

Enrico Zorzetto,
On behalf of all authors.

Reviewer #1

This study describes the implementation of a new snow albedo scheme in the GLASS land surface component of the GFDL climate model, accounting for the effects of light-absorbing particles (LAPs).  It then evaluates the simulation of snow mass, snow depth, and surface albedo at several sites contributing to the SnowMIP effort.  Overall, this study represents an advance in scientific capabilities of the GFDL climate model.  I found it particularly useful that the model was run in "single point" mode for comparisons with the SnowMIP sites.  The quantification of the reduction in number of snow cover days due to the presence of LAPs was also useful, though it would be helpful to also include an evaluation of LAP concentrations in snow, compared with observations, to help understand how realistic the simulated LAP-induced albedo effect is.  The manuscript is generally well-written and well-organized.  Aside from one major suggestion, I have only minor comments.

We thank the referee for the  insightful comments on our work. Please find below a point-by-point response to the comments on the manuscript. We believe addressing these comments did significantly improve the manuscript with respect to our first submission. In particular, to address the major comment in this review we have extended our analysis by adding a comparison of our model results to dust-on-snow observations.

Major comments:

The new model prognoses the mixing ratios of dust and black carbon in surface snow, and I believe that concentrations of these particles have been measured at some of the SnowMIP sites used in the evaluation, such as Senator Beck and Col du Port.  It would be quite helpful to know how the simulated mixing ratios compare with observations, as this would inform on potential sources of bias in the simulated SWE evolution throughout the seasons.  Such an evaluation could suggest, for example, that biases in impurity concentrations are responsible for SWE biases, or conversely if the particle mixing ratios appear realistic, that there are other problems with the snow model.

We agree that comparing our model results to measured concentrations of LAPs in snow is a helpful addition to our analysis. For the Col de Porte site, we have already compared our results with spectral measurements from Dumont et al., 2017, which estimate an equivalent concentration of LAPs at the snow surface for one snow season (Figure 3).

For the Senator Beck Basin (snb) and Swamp Angel (swa) sites, we have now added an additional comparison with i) a dataset for the the 2013 snow season which includes mineral dust and black carbon concentration data published by Skiles and Painter (2017), and ii) a multi-year dataset of end-of-season dust concentrations in snow (Skiles and Painter, 2015).

We have added to the manuscript a description of these additional datasets used in the analysis as follows (Text added at line 309 of the revised manuscript):

*In addition, for validation over the Senator Beck Basin (snb) site, we employ a dataset of dust-in-snow observations collected by Skiles and Painter (2015, 2017). These include end-of-year concentrations of dust within the snow for the years 2005-2012 (Skiles and Painter, 2015), and a sequence of measurements characterizing the seasonal evolution of MD and BC concentration in snowpack for the year 2013 (Skiles and Painter, 2017). Measured concentration values in this dataset correspond to average concentrations within the uppermost 30 cm of the snowpack. In the following, we compare these values to average modelled concentrations within the entire snowpack, and to the predicted concentration values over the snow near-surface layer.*

We have now performed an evaluation of our model simulations based on these additional measurements. We have included two additional figures to the manuscript reporting the results of this new analysis as follows (the new figure 4 and figure 5 that have been added to the manuscript are reported below). At line 366:

[revised manuscript text omitted]

Section 2.6: It was not apparent to me how/if the albedo of the ground underlying snow and snow thickness affect the snow albedo calculation. Is the snow assumed to be optically "semi-infinite" regardless of snowpack thickness? If so, this would cause a high bias in the albedo of thin snowpack, and it should be acknowledged.

We agree that this limitation of the model should be discussed in the manuscript. The parameterization is for semi-infinite snowpack and thus can give rise to errors in case of thin snowpacks. We now clarify this point in the revised manuscript instead of just referring to Zorzetto et al., 2024 and to He et al., 2018 for details on the albedo parameterization used.

We now explicitly specify in the revised manuscript that in the case of thin snowpacks, a snow area fraction is computed and used for albedo calculations. At line 265:

265     Note that the albedo parameterization employed here based on the work of Dang et al. (2015) and He et al. (2018a) is derived for a semi-infinite snowpack, and thus in general can lead to biased albedo estimate for thin snowpack. In the case of this snowpack, the model computes a fractional snow cover $f_{snow}$ based on snow depth as follows

$$f_{snow} = \frac{h_s}{h_s + h_{s,c}} \qquad (7)$$

with $h_s$ the snowpack depth, and $h_{s,c} = 0.0167$ m. In case of fractional snow cover, surface albedo is computed as a weighted 270    spatial average of snow and snow-free substrate optical properties.

In the discussion we now explicitly mention this issue by stating that (line 467):

*"Furthermore, the model could be improved by using more detailed optical models such as the Two-stream Analytical Radiative TransfEr in Snow (TARTES) model (Libois et al, 2013) or the Snow, Ice, and Aerosol Radiative (SNICAR) model (Flanner and Zender, 2005), which are not limited to the case of semi-infinite snowpack as is the case for the albedo parameterization used here."*

Minor comments:

1. line 55: "Black carbon has the largest absorption..." -> Black carbon has the largest absorption *per unit mass* ..."

We agree with the comment, and have revised the text as suggested.

2. line 119: "sol-snow" -> "soil-snow"

We agree with the comment, and have revised the text as suggested.

3. Lines 132-146, section 2.2: Overall, this is a helpful summary. Briefly, though, could you please also list the maximum number of snow layers allowed in this model, along with maximum/minimum layer thicknesses (especially near the top)? This info is probably available in the companion paper, but it would be helpful to include it here, too.

We now discuss more in detail the snow vertical layering scheme used in GLASS. At line 146 of the revised manuscript we have added the following:

*"The vertical structure of the snowpack consists of a dynamic number $n_L$ of snow layers. New layers are created on top of the existing snowpack following snowfall events of large enough magnitude, so that the vertical layering structure preserves snow physical properties in each layer. Depending on snowfall rate, up to 5 new snow layers can be created during a single model time step.*

*The vertical layers are also updated based on computational considerations. At each time step, the snow vertical structure is compared to an optimal vertical discretization defined for each given snow depth. If the layers are too coarse or too thin for a given snow depth, the layers undergo splitting or merging. In the current configuration, the optimal thickness of the uppermost snow layer is set to 3 cm , and each snow layer optimal thickness is set to 1.5 that of the layer immediately above it, so that in general the model allows for thinner layers closer to the surface, while within the snowpack layer thickness increases with depth. The model does not prescribe a maximum number of layers, while if snow is present the minimum number of layers is 3, as required for numerical solution of mass and energy vertical balance equations.*

*These operations are designed in order to strike a trade-off between computational cost and vertical detail, to satisfy requirements of numerical efficiency (to avoid too large number of layers) and to ensure a proper description of the snowpack vertical structure (too coarse a vertical discretization would hinder the representation of some physical processes, such as the vertical heat diffusion). If two snow layers are characterized by values of density, optical diameter or impurities content which are too different, merging of the two layers is not permitted in order to preserve the vertical heterogeneity of the snowpack. "*

4. lines 158-164: Would it be accurate to state that the mixing ratio of LAPs within precipitation is held constant throughout the month? Also, is there any interpolation between the months, or does the mixing ratio change abruptly on the first day of the month?

The mixing ratio is kept constant within each month and then indeed changes abruptly to the next value. While this is not ideal, we believe it was the simplest approach given the available data. We now clarify this point at line 189 of the revised manuscript:

*"Since we force the snow model with in-situ observations, we adopt the following strategy to estimate wet deposition fluxes for each snowfall or rainfall event: We first compute the monthly average concentration of each LAP specie in the precipitation (as the mass ratio of monthly average wet deposition to monthly precipitation, in [ppm]). We then assign in each model time step the total amount of wet-deposited LAPs as proportional to the rainfall and/or snowfall rate for that time step, so that the flux of tracer i due to liquid and solid precipitation is respectively $c_{l,i} f_l$ and $c_{s,i} f_s$. Note that based on this procedure the mixing ratio of LAPs in precipitation is constant during each month, and exhibits step changes across months. For the purpose of this study, we assume that rainfall and snowfall carry the same concentration of LAPs, and neglect any possible dependence of deposition fluxes e.g., on precipitation intensity."*

5. lines 158-168: Related, is there an interactive, coupled version of the atmosphere and land models, where prognosed aerosol deposition is coupled with GLASS each timestep? (Or, are there plans to extend this modeling framework to the coupled model?)

While the model was tested here in a "land-only" configuration, it was indeed designed for coupled Earth System Model simulations. While not included with the software package distributed with this publication, a coupled version of the model is currently under development and its evaluation will be the object of future research. The current land-only model version is ready for such coupled runs, with the caveat that deposition fluxes for each LAP species will need to be provided by the atmospheric model. In this setup, the computation of the monthly constant mixing ratio of LAPs in precipitation performed here (see our response to the previous comment #4) will no longer be needed as both liquid/frozen water fluxes and LAP deposition fluxes will be provided by the atmospheric model. We plan to test this coupled configuration in future research. We have now added a brief note about this planned research in the discussion section of the revised manuscript (line 480):

*"While the model was here tested in a land-only configuration forced by an offline atmosphere, LM4.1 and GLASS are designed as components of an Earth System Model and are thus used for coupled simulations with an atmospheric model. In this coupled model configuration, currently under development, the computation of the monthly constant mixing ratio of LAPs in precipitation will no longer be needed as both liquid/frozen water fluxes and LAP deposition fluxes will be provided by the same atmospheric model. This would further reduce one of the sources of uncertainty here, due to the coarse temporal resolution of LAP deposition fluxes.*

*"*

6. line 193: Is the same scavenging ratio assumed for hydrophilic and hydrophobic BC? It seems that the scavenging ratio should be larger for hydrophilic BC, as specified by Flanner et al (2007).

In the current configuration, we consider a single species of BC which includes both hydrophobic and hydrophilic components, with total BC deposition fluxes provided by in the input dataset. We agree that scavenging ratios as well as optical properties should distinguish between the two. We now explicitly state that this is the case for the current model configuration, and note that it can be extended to treat hydrophilic and hydrophobic BC separately as long as scavenging coefficients and optical properties are available. At line 173 of the revised manuscript, we add the following:

*"These quantities ( $w_{IM,i,k}$ and $w_{EM,i,k}$ ) in [kg $m^{-2}$] are tracked for each tracer species $i$, so that in our current application we have 6 types of LAP in each layer (IM and EM, for each species: BC, OC and MD). The model considers a single LAP size distribution for mineral dust,*

*without tracking separately dust particles of different sizes. Similarly, the current model considers a single BC species and does not distinguish between hydrophobic and hydrophilic components. This is a limitation as hydrophobic and hydrophilic BC species have different optical properties and scavenging coefficients (Flanner et al., 2007). However, we note that the model can in principle be extended to track multiple BC species (or multiple dust size bins) as long as their optical properties and scavenging coefficients are known.”*

7. line 196-197: "... snow properties are averaged over a near surface layer of thickness set equal to up to 3cm." - Is this simply the top thermodynamic snow layer in GLASS, or is this a weighted average of multiple snow layers?  When is it less than 3cm?  Please elaborate a bit on this scheme.

This surface layer is always 3 cm deep, or as deep as the entire snow depth, whatever is thinner. If the top snow layers are thinner than this value, a weighted average over multiple snow layers is performed. We now clarify this feature of the model at line 228 of the revised manuscript:

*“In this work, the snow surface albedo is computed based on snow properties (optical diameter and shape) and on the concentration of LAPs near the snowpack surface. In this section, snow properties are averaged over a near-surface layer of thickness set equal to up to 3 cm. If the snowpack is thinner than 3 cm, the near-surface layer includes the entire snow depth. If the upper snow layers are thinner than 3 cm, the near-surface snow properties are computed as weighted average across snow layers of the snow properties in each layer, up to a 3 cm depth.“*

8. lines 215-232 (Section 2.7): Do LAPs influence albedo in both spectral bands, or only the visible band?  Line 221 mentions "as a function of spectral band", but line 230 lists only single absorption cross-sections for each type of LAP.  Are these absorption cross-sections for the visible band only, and if so, what is assumed for the near-IR band?  Also, what are the spectral intervals of the two bands used in this model?   (Often they are separated at 700 nm)

The LAPs affect only snow optical properties in the visible range. We now explicit mention this at line 242 of the revised manuscript:

*“This effect is present only for the visible band (b=VIS) as described in Section 2.7, while $\Delta\alpha_{NIR} = 0$.”*

Furthermore, we now clarify eq. (6) and we specify that this albedo reduction only applies to the visible band (line 271):

*“The effect of LAPs is accounted for using the parameterization by He et al., (2018), in which the albedo reduction in the visible range is obtained as..”*

The separation between bands is indeed at 700 nm; We now explicitly state this in the revised manuscript as follows (line 232):

*"In GLASS, the shortwave radiative balance is resolved for two bands, visible (VIS) and near infrared (NIR), separated at 700 nm. Based on the work of Dang et al., (2015) and He et al., (2018), in GLASS the snow surface albedo for each band (b=VIS or b=NIR) is expressed as a function of snow grain effective radius…"*

9. lines 290-295: The assumption of constant LAP mixing ratios within precipitation throughout the month could also explain some of this discrepancy.

We agree this is indeed likely the case, and we now discuss this in the revised manuscript as follows (line 358):

*"The concentration of LAPs in the near surface snow layer is also well captured by the model (Figure 3B), considering that forcing values are obtained from an atmospheric model climatology dataset. For this reason, we do not expect the model to closely match variations in $c_{eq,ns}$ throughout the entire snow season. The order of magnitude of observed LAP concentration is comparable with simulations, but intra-seasonal variations are not captured by the model. This could partially be due to the fact that the mixing ratio of LAPs in precipitation is assumed constant for each month. During spring, the snow ablation phase is characterized by a sharp increase in LAP concentration driven by the combined effect of sublimation and melt. During this phase, the increase in $c_{eq,ns}$ is overall well represented in the model."*

10. Figure 3: Please explain this "retrieval parameter" for the different observational curves, perhaps in the text.

We now explicitly discuss the meaning of the retrieval parameter in the revised manuscript. At line 303:

*"Additionally, spectral measurements were carried out in the snow year 2013-2014 at the Col de Porte site (Dumont et al, 2017), which were then used to estimate snow specific surface area (SSA) and concentration of LAPs. Dumont et al;, (2017) used a theoretical spectral model to infer snow surface properties from a set of observed spectra. To quantify the uncertainty and artifacts in the measurements, a scaling factor a was used to relate theoretical and observed spectra. To quantify the uncertainty of these retrievals, here we use the snow surface properties computed by Dumont et al;, (2017) for three values of this scaling factor, corresponding to the $25^{th}$ (a=0.920), $50^{th}$ (a=0.943), and $75^{th}$ (a=0.964) quantiles of its distribution. "*

11. Please include a table describing the acronyms of the SnowMIP sites (clp, snb, etc), including the long names and locations of the sites.

We have added to the manuscript the following table summarizing the main characteristics of the study sites:

**Table 1.** Characteristics of the experimental sites used for model validation.

| Station | Name | Years | Latitude | Longitude | Elevation [m] | Climate type |
|---|---|---|---|---|---|---|
| Col de Porte, FR | *cdp* | 1994-2014 | 45.30 N | 5.77 E | 1325 | Alpine |
| Reynolds Mountain East., USA | *rme* | 1988-2008 | 43.06 N | 116.75 W | 2060 | Alpine |
| Senator Beck, USA | *snb* | 2005-2015 | 37.91 N | 107.73 W | 3714 | Alpine |
| Swamp Angel, USA | *swa* | 2005-2015 | 37.91 N | 107.71 W | 3371 | Alpine |
| Weissfluhjoch, CH | *wfj* | 1996-2016 | 46.83 N | 9.81 E | 2540 | Alpine |
| Sapporo, JP | *sap* | 2005-2015 | 43.08 N | 141.34 E | 15 | Maritime |
| Sodankyla, FI | *sod* | 2007-2014 | 67.37 N | 26.63 E | 179 | Arctic |

12. Section 4.2: As mentioned under Major Comments, this analysis would be augmented with an evaluation of the impurity amount (mixing ratio) in snow.

In addition to the comparison with spectral measurements at Col De Porte, we have now performed a comparison for the Senator Back Basin and Swamp Angel sites using data published by Skiles and Painter (2015) and Skiles and Painter (2017).

See our detailed response to this comment under our response to "major comments".

13. Figures 4-6: Which "single year" was simulated and observed at each site? (And were they the same?) Please explain and/or include this information in the table of sites requested two comments above

When reporting a single year of our results (Figs 4, 5, and 6) we have selected the same snow year for all sites (year 2013-2014) except for the *"rme"* site, where we show the 2003-2004 year instead because no observations were available at the site for the 2013-2014 year. The plots x-axis in the original manuscript already report the dates for all sites, but we now mention this explicitly in the revised manuscript (in each figure's caption) in order to emphasize this difference and to avoid any confusion. This issue is further clarified by the newly introduced Table 1 (see response to previous comment #11), where we report the start and end year of the observational record available for each study site.

14. Overall, the grammar is good, but there are numerous instances of minor issues that should be fixed prior to publication. Lines 79-80 demonstrate just one example of a sentence that needs to be cleaned up.

We checked the grammar and style throughout the manuscript correcting multiple issues. For example, the sentence at lines 79-80 was rephrased as follows:

*"Therefore, understanding to what extent the representation of LAP-on-snow processes contributes to the uncertainty in snow predictions from regional and global modeling efforts is a key scientific question which in the last decade has received increasing attention in the Earth System Modelling community (Qian et al., 2015, Réveillet et al., 2022; Hao et al., 2023b)".*

Reviewer #2

The authors expanded the GFDL LM4.1 snowpack processes to include LAPs effects and evaluate the simulated snow albedo across different measurement sites. They found that the resulting snow model with LAP-aware snow reflectivity show a good agreement with measurements of broadband albedo and seasonal SWE over the study sites. They further evaluated the number of snow-days lost due to the deposition of dust and carbonaceous aerosols, which ranges between 5 and 24 days depending on locations. This work is an important improvement for the GFDL snowpack model, which could provide better simulations for future coupled climate runs. Overall, the manuscript is well organized. I have a few specific comments/suggestions for the authors to consider.

We thank the reviewer for providing insightful comments on our work. Please find below a point-by-point response to the comments on the manuscript. We believe addressing these comments significantly improved the manuscript with respect to our first submission.

Specific comments:

1. Is there any canopy snow process included in LM4.1? A brief description would be useful.

   The land model GFDL LM4.1 in general resolves snow intercepted by canopy layers, computing its full mass and energy balance. The albedo of leaves is a weighted average which considers the amount of fractional leaf area covered by snow. However, in the current application vegetation is not considered for the sites. In this work, we have limited our analysis to experimental sites with little to no vegetation cover, in order to focus our attention on the effect of impurities rather than on the complex snow-vegetation interactions. Therefore, we have turned off vegetation in the model, which effectively is not allowed to grow throughout the simulation. We have expanded the discussion on this model setting in the manuscript, and we now note that further investigations focusing on forested sites would be an important topic of future research. To clarify this point, we have added the following text to the manuscript (at line 125 of the revised manuscript):

*"In LM4.1, vegetation is represented by a set of plant cohorts which evolve dynamically. Multi-layer vegetation canopies interact with the surface via multiple processes, including turbulent exchange of mass and energy, the transfer of longwave and shortwave radiation, and the interception of liquid and frozen precipitation. For additional details, the reader is referred to Shevliakova et al., (2024). In this application, we have decided to limit our analysis to sites with little to no vegetation in order to focus our attention on snow and LAP deposition processes. Therefore, vegetation is turned off and canopy layers are not present in the model simulations, following the experimental setup used in Zorzetto et al. (2024) for sites with little or no vegetation."*

However, we believe that further validation of the model proposed here over forested areas would be an important and informative future research objective. We now mention this issue also in the discussion section of the revised manuscript at line 475:

*"Furthermore, the current application was limited to sites with little to no vegetation. We believe further investigation of model performance and of the effects of LAPs for snowpacks in forested areas would be an important addition to this line of research."*

2. How is the snowpack liquid water treated? Is there a liquid water holding capacity parameter prescribed? How important is it to assign different inter-layer snowmelt water scavenging coefficients for IM vs EM LAPs?

We agree that the treatment of liquid water flow was not discussed in the original paper, and it is a relevant process for the transport of LAPs. We now include a detailed description of these processes instead of just referring the reader to the model description paper (Zorzetto et al., 2024). We now include the following discussion at line 162:

*"At each model time step, the full energy and water mass balance of the snowpack is solved using an implicit numerical formulation, which is required for land-atmosphere coupled model runs with relatively coarse time step (30 min) for which the GLASS was designed. After performing the vertical heat balance of the snowpack, the temperature change and change of phase is evaluated for all snow layers. The vertical balance of liquid water is then evaluated throughout the snowpack, with liquid precipitation providing the upper boundary condition. The snow density is used to evaluate the pore space available for liquid water in each snow layer. Following Vionnet et al., (2012), the liquid water holding capacity for a snow layer k with thickness $\Delta z_k$ and local solid-phase density $\rho_{s,k}$ is given by*

$$W_{liq,max,k} = 0.05 \rho_w \Delta z_k \left( 1 - \frac{\rho_{s,k}}{\rho_i} \right)$$

*with $\rho_i$ the density of ice, and $\rho_w$ that of liquid water. For a more detailed description, see Zorzetto et al., (2024)."*

With respect to the second point raised in this comment, we now clarify that we are using a single scavenging coefficient for IM and EM LAPs, and in the revised manuscript we now discuss potential extensions of the model with more realistic scavenging parameterizations. We have added following clarification at line 453 of the revised manuscript:

*"The modelled LAP concentration directly depends on how LAP scavenging processes in the snowpack are represented in the snow model. In the current GLASS formulation, scavenging coefficients are constant across snow layers, and the three LAP species considered here do not account for the different behavior between e.g., hydrophobic and hydrophylic carbon components. Furthermore, scavenging coefficients do not depend on the mixing ratio of impurities (i.e., internally or externally mixed). Future extensions of the model could include more realistic scavenging parameterizations depending on the LAP mixing state."*

3. How did the authors assign dry and wet deposited LAPs to IM or EM within the snow?

Dry deposition contributes to EM impurities in the snowpack, while wet deposition (from both liquid and frozen precipitation) contributes to IM LAPs. We now clarify this feature of the model in the revised manuscript. The addition at line 181 of the revised manuscript reads:

*"The mass of LAPs added to the snowpack by dry deposition is assumed to be externally mixed (EM), while LAPs deposited as wet deposition either due to liquid or frozen precipitation contribute to IM LAPs within the snowpack. In the model, the state of mixing of LAPs within the snow (IM or EM) does not change as a consequence of melt and freeze cycles occurring in a snow layer."*

4. How many dust size bins are considered and what are they?

We have considered a single bin size with a single absorption value for dust. This is clearly a limitation, but can be extended in future studies provided that dust bins sizes and their respective optical properties are available as model input data. We now explicitly mention this in the manuscript (line 176):

*"The model considers a single LAP size distribution for mineral dust, without tracking separately dust particles of different sizes. "*

5. Does melt-freeze of snow change the IM or EM status of LAPs?

As already mentioned in our response to comment #3, In the current model configuration, melt/freeze cycles do not change the IM/EM status of LAPs within the snowpack. We now clarify this point in the revised manuscript (line 181):

*"The mass of LAPs added to the snowpack by dry deposition is assumed to be externally mixed (EM), while LAPs deposited as wet deposition either due to liquid or frozen precipitation*

*contribute to IM LAPs within the snowpack. In the model, the state of mixing of LAPs within the snow (IM or EM) does not change as a consequence of melt and freeze cycles occurring in a snow layer."*

6.  Is internal heating within the snowpack column due to light absorption considered?

Yes, as mentioned in the original manuscript we do consider the penetration of light within the snowpack and related heating rates. We discuss this feature of the model in the revised manuscript as follows (line 258):

For thick enough snowpack (snow depth $> 0.02$ m), solar radiation penetrates within the snowpack and absorbed radiation is distributed exponentially

$$\quad Q_s(z) = \sum_{b=1}^{2} (1 - \alpha_b) R_{s,b} e^{-\beta_b z} \tag{6}$$

where for each band $b$ $R_{s,b}$ is the downward shortwave radiation at the surface, and $\beta_b$ describes the penetration of light within the snowpack. The extinction coefficients for visible and near infrared light are estimated as in Jordan (1991) and Shrestha et al. (2010): $\beta_{NIR} = 400$, and $\beta_{VIS} = 0.003759 \, \rho \, d_{opt}^{-0.5}$, with density and optical diameter averaged over the near-surface layer of the snowpack, up to a maximum depth of 3 cm.

7.  The snow albedo parameterization from Dang et al. 2015 and He et al. 2018 is for semi-infinite snowpack, which may lead to uncertainties in the albedo calculation here. It will be good to clarify this and briefly discuss this.

We agree that this limitation of the model should be discussed in the manuscript. The snow albedo parameterization employed is indeed for semi-infinite snowpack and thus can give rise to errors in the case of thin snowpacks. We now explicitly discuss this point in the revised manuscript, instead of just referring to Zorzetto et al. (2024).

Furthermore, in the revised manuscript we explicitly describe how thin snowpacks are treated in the model. In particular, we mention that in the case of thin snowpacks, a snow area fraction is computed and used for albedo calculations. We have added the following to the model description (line 265):

265      Note that the albedo parameterization employed here based on the work of Dang et al. (2015) and He et al. (2018a) is derived for a semi-infinite snowpack, and thus in general can lead to biased albedo estimate for thin snowpack. In the case of this snowpack, the model computes a fractional snow cover $f_{snow}$ based on snow depth as follows

$$f_{snow} = \frac{h_s}{h_s + h_{s,c}} \tag{7}$$

with $h_s$ the snowpack depth, and $h_{s,c} = 0.0167$ m. In case of fractional snow cover, surface albedo is computed as a weighted
270     spatial average of snow and snow-free substrate optical properties.

In the discussion we now further discuss this issue by stating that (line 467):

*"Furthermore, the model could be improved by using more detailed optical models such as the Two-stream Analytical Radiative TransfEr in Snow (TARTES) model (Libois et al, 2013) or the Snow, Ice, and Aerosol Radiative (SNICAR) model (Flanner and Zender, 2005), which are not limited to the case of semi-infinite snowpack as is the case for the albedo parameterization used here. "*

8. How did the authors use the snow grain shape parameters to compute the optical diameter?

Prompted by this comment, we have expanded the description of how snow grain size and shape are used to compute the snow albedo. At line 249 of the revised manuscript, we have added the following discussion:

*"The snow albedo parameterization used here explicitly accounts for the effects of snow grain size (through the snow grain effective radius) and shape on its optical properties. He et al., (2018) introduced eq. (3) and provided the set of parameters $b_0, b_1$ , and $b_2$ tabulated for four different snow grain shapes (sphere, spheroid, hexagon, and Koch snowflake). In GLASS, snow microphysics in each snow layer is parameterized by two parameters (snow sphericity and dendricity) which evolve in time due to the combined effect of dry and wet snow metamorphic processes, as well as due to wind effects (Zorzetto et al., 2024). The coefficients used in eq. (3) are selected at each time step based on snow shape properties in the near-surface snow layer: High-dendricity snow ($\delta_{ns}$ > 0.5) is idealized as a collection of Koch snowflakes. Snow with lower dendricity parameter is considered as a collection of spheres (if sphericity parameter $s_{p,ns}$ > 0.8), spheroids (if 0.8 > $s_{p,ns}$ > 0.2), or hexagonal crystals (if $s_{p,ns}$ < 0.2)."*

9. How is the alpha_b in equation (5) computed? What is the physical meaning of this parameter?

Here alpha_b is the snow surface albedo, which is computed according to eq. (2) based on physical properties on the snowpack (i.e., grain size and shape). We believe the source of this misunderstanding has been removed in the revised manuscript: We have now updated the notation used in the paper, and we are now using the suffix b for snow surface albedo also in eq. (2) where it was not used in the original submission. Therefore, we now clarify the dependence of albedo on the shortwave band (b = VIS or NIR for visible or near infrared band, respectively) and the notation used is consistent throughout the revised manuscript.

10. For daily average of albedo, did the authors use downward solar radiation as the weights?

Yes, we indeed obtained dimensional quantities from the model output (downward and upward radiation fluxes) and computed their ratio at the daily time scale at which results are displayed.

We now explicitly mention this in the manuscript to clarify the procedure used in the analysis. At line 333 of the revised manuscript we add the following:

*"In order to compare modelled and observed surface albedo, modelled upward and downward shortwave radiation fluxes are averaged daily to correspond to observations"*

11. Figure 2A: Why does the model have a consistently high snow albedo than observations after 2014 May (which seems to be a snow-free period)? Also, since there is no snow after May 2014, why is the snow albedo from both observation and model not zero?

We now clarify that the quantity shown both in Figure 2A and Figure 5 of the original manuscript is the surface albedo, averaged at the daily time scale, which does not always correspond to the snow albedo. This is the same quantity provided in the observational datasets used for comparison. We now clarify this in the figure captions to explain why albedo values are shown even for snow-free periods. Furthermore, in the revised paper we now mention that for thin snowpacks the model computes an effective snow fractional area and therefore surface albedo is effectively based on a weighted spatial average of snow and snow-free substrate optical properties. At line 265 of the revised manuscript we have now added the following clarification:

265    Note that the albedo parameterization employed here based on the work of Dang et al. (2015) and He et al. (2018a) is derived for a semi-infinite snowpack, and thus in general can lead to biased albedo estimate for thin snowpack. In the case of this snowpack, the model computes a fractional snow cover $f_{snow}$ based on snow depth as follows

$$f_{snow} = \frac{h_s}{h_s + h_{s,c}} \tag{7}$$

with $h_s$ the snowpack depth, and $h_{s,c} = 0.0167$ m. In case of fractional snow cover, surface albedo is computed as a weighted
270    spatial average of snow and snow-free substrate optical properties.

12. Section 4: the description of the results is too qualitative. Please include some quantitative numbers when presenting the results. Also, more physical explanations/insights could be added to the results. For example, why does the model not capture the variation of the daily albedo variation over most sites (Figure 4) and why does the model results show systematic overestimate/underestimate in some sites (Figure 5).

We have now added to the results section of the manuscript a more quantitative discussion of the discrepancies between modeled and observed snowpack properties as suggested in this comment. At line 404 of the revised manuscript we now discuss the bias in SWE:

*We note that, for the sites which exhibit the largest SWE discrepancies between model output and observations, the effect of LAPs predicted by the model does not appear to be the primary reason for model biases. In particular, accounting for LAPs does not appreciably change the seasonal peak SWE, which the model underestimates by about 22% at two of the sites (swa and wfj) and by about 34% at sap.*

For albedo, at line 392:

*"During the accumulation phase, some underestimation of daily albedo by the model can be observed at some of the sites (snb, sap, and wfj). It is worth noting that some of the sites where the model exhibits the largest SWE underestimations correspond to sites where a negative bias in snow surface albedo is also reported (e.g., at wfj and especially at sap, which is the site characterized by the largest SWE and albedo underestimation in our dataset) so that surface albedo appears the primary source of the SWE bias."*

And at line 397 we have added:

*Furthermore, the temporal variability of modelled albedo is generally smaller than the observed one. This can be the result of effects due to the snow surface grain properties, and can also depend on how the solar angle is included the albedo parameterization through the coefficient $\Phi_b(\mu)$.*

Now at line 459 we have added a comparison with observed concentration of LAPs:

*"Uncertainty in modelled LAP concentrations and snow optical properties also depends on the dataset of input LAP deposition fluxes used here. While the other atmospheric variables used to force the land model consists of in-situ observations, LAP deposition data are obtained from a reanalysis dataset. Therefore, LAP fluxes used here are coarser in space and time and may not be fully representative of the local deposition flux at the sites. Furthermore, here we assume that dust absorption properties are constant globally, while in general they do depend on dust mineralogy, which is spatially heterogeneous. The comparison with LAP concentration observations at two of the sites helped us constrain these sources of uncertainty and showed that while modelled LAP concentrations exhibit discrepancies with observations, overall the order of magnitude and seasonal trend throughout a snow season are reproduced by the model."*

13. Figure 7: It seems that the model SWE bias is dominated by other model snow or forcing processes instead of LAP effects. This may be worth some discussion.

We agree and now discuss this issue at line 403 of the revised manuscript as follows:

*"SWE model predictions show a good fit to observations at most SnowMIP sites (Figure 7). However, for some stations overestimation (rme, snb) or underestimation of SWE (swa, sap, wfj) are observed for the specific snow year examined. We note that, for the sites which exhibit the largest SWE discrepancies between model output and observations, the effect of LAPs predicted by the model does not appear to be the primary reason for model biases. In particular, accounting for LAPs does not appreciably change the seasonal peak SWE, which the model underestimates by about 22% at two of the sites (swa and wfj) and by about 34% at sap.*

14. I would suggest adding a subsection for uncertainty discussion. Some of the uncertainties involved in the model are mentioned in my earlier comments. A few key uncertainty factors that are worth discussing: (1) snow grain shape and size evolution, (2) using Eq.8 to combine different LAPs, (3) aerosol deposition flux, (4) missing snowpack processes, (5) LAP meltwater scavenging, etc.

We agree. Given the number of processes involved we believe that a complete quantitative assessment of model uncertainty is beyond the scope of the present work. However, discussing these sources of uncertainty is important and helps to assess our results.

We have added a subsection to the Discussion section entitled *"Sources of uncertainty and model limitations"* where we expand our discussion of model limitations and sources of uncertainty. At line 452 of the revised manuscript we have added the following discussion subsection:

*"*

*5.1 Sources of uncertainty and model limitations*

*The modelled LAP concentration directly depends on how scavenging processes of LAPs in the snowpack are represented in the snow model. In the current GLASS formulation, scavenging coefficients are constant across snow layers, and the three LAPs species considered here do not account for the different behavior between e.g., hydrophobic and hydrophylic carbon components. Furthermore, scavenging coefficients do not depend on the mixing state of impurities (i.e., internally or externally mixed particles). Future extension of the model could include more realistic scavenging parameterization depending on the LAP mixing state.*

[revised manuscript text omitted]

Libois, Q., et al. "Influence of grain shape on light penetration in snow." *The Cryosphere* 7.6 (2013): 1803-1818.

Flanner, Mark G., and Charles S. Zender. "Snowpack radiative heating: Influence on Tibetan Plateau climate." *Geophysical research letters* 32.6 (2005).

Shevliakova, E., et al. "The land component LM4. 1 of the GFDL Earth System Model ESM4. 1: Model description and characteristics of land surface climate and carbon cycling in the historical simulation." *Journal of Advances in Modeling Earth Systems* 16.5 (2024): e2023MS003922.

Vionnet, Vincent, et al. "The detailed snowpack scheme Crocus and its implementation in SURFEX v7. 2." *Geoscientific model development* 5.3 (2012): 773-791.

Flanner, Mark G., et al. "Present‐day climate forcing and response from black carbon in snow." *Journal of Geophysical Research: Atmospheres* 112.D11 (2007).